

# A coupled hydrological and hydrodynamic modelling approach for estimating rainfall thresholds of debris-flow occurrence

**Zhen Lei Wei[a, c], Yue Quan Shang[b], Qiu Hua Liang[c*], Xi Lin Xia[d]**

a State Key Laboratory of Geohazard Prevention and Geoenvironment Protection, Chengdu University of

Technology, Chengdu, 610059, China

b College of Civil Engineering and Architecture, Zhejiang University, Hangzhou 310058, China

c School of Architecture, Building and Civil Engineering, Loughborough University, Loughborough LE11 3TU, UK

d School of Engineering, University of Birmingham, Birmingham B15 2TT, UK

* Corresponding author: Qiuhua Liang, q.liang@lboro.ac.uk

## Abstract

Rainfall-induced hydrological processes and surface water flow hydrodynamics may play a key role in initiating debris flows. In this study, a new framework based on an integrated hydrological and hydrodynamic model is proposed to estimate the Intensity-Duration (I-D) rainfall thresholds that trigger debris flows. In the new framework, intensity-duration-frequency (IDF) analysis is carried out to generate design rainfall to drive the integrated models to calculate grid-based hydrodynamic indices (i.e. unit-width discharge). The hydrodynamic indices are subsequently compared with hydrodynamic thresholds to indicate the occurrence of debris flows and derive rainfall thresholds through the introduction of a zone threshold. The capability of the new framework in predicting the occurrence of debris flows is verified and confirmed by application to a small catchment in Zhejiang Province, China, where observed hydrological data are available. Compared with the traditional statistical approaches to derive Intensity-Duration (I-D) thresholds, the current physically-based framework can effectively take into account the hydrological processes controlled by meteorological conditions and spatial topographic properties, making it more suitable for application in ungauged catchments where historical debris flow data is lacking.

**Keywords:** debris flow; hydrological model; hydrodynamic model; rainfall threshold; hydrodynamic indicator




## 1. Introduction

30      As a common type of natural hazard in mountainous areas, debris flows usually consist of a mix of rocks, mud, water, and air (Hürlimann et al., 2019). The velocity and impact force of a debris flow could be tremendous, imposing a serious threat to the people, property, and infrastructure systems in the affected areas. It is important to establish early warning systems to enhance the preparedness of at-risk communities to reduce potential impact. Early warning may be achieved through reliable estimation of rainfall thresholds linked to the occurrence of debris flows.

     Considering the hydrological interaction between debris flows and rainfall, two types of initiation mechanisms have been identified: 1) debris flows initiated by landslides (Iverson et al., 1997), and 2) debris flows triggered by runoff (Kean et al., 2013). A landslide-triggered debris flow often involves loose soils or materials overlying the bedrock on a steep slope following a landslide. When rainfall-induced infiltration increases the saturation level of the soil (initially unsaturated) above the infiltration front or forms a perched water table in the superficial soil layers, the loose soil may become unstable and develop into a debris flow (Berti and Simoni, 2010; Godt et al., 2009). For runoff-generated debris flows, different initiation mechanisms are recognized, which may be related to grain-by-grain erosion, mass failure, bank failure and the so-called 'firehose' effect (Gregoretti and Dalla Fontana, 2008). The current study will focus on runoff-generated debris flows.

     Previous studies have demonstrated that three key factors may contribute to the triggering of debris flows, including steep slope, availability of sediment, and input water flow (Mcguire et al., 2017; Coe et al., 2008; Imaizumi et al., 2006; Hürlimann et al., 2014; Berti and Simoni 2005). For a catchment prone to debris flows, the water inflow that triggers debris flows usually varies rapidly across the temporal and spatial scales (Gregoretti and Dalla Fontana 2008; Cannon et al., 2008). Rainfall provides the primary source of water inflow and the strong correlation between debris flow initiation and rainfall conditions has been confirmed in many existing studies (Berti et al., 2020). Estimation of the rainfall conditions triggering debris flows, i.e. rainfall thresholds, has become a widely used approach to support early warning (Wei et al., 2017; 2018; Guzzetti et al., 2008).

     At present, the most commonly used rainfall thresholds of debris flows are derived from intensity-duration (I-D) curves due to the simple calculation process and availability of influencing factors (Guzzetti et al., 2008). The traditional I-D rainfall thresholds are mostly generated by analyzing the historical data of debris flow occurrence, and the intensity and duration of the triggering rainfall events using statistical approaches (Guo et al., 2016; Ma et al., 2016; Staley et al., 2013). The generation of these statistical I-D rainfall thresholds relies on the availability of rich datasets of rainfall events that have triggered debris flows. However, debris flow is a rare event with low occurrence frequency, making it challenging to collect high-quality observation data, especially for a specific gully or catchment. Furthermore, due to the spatial variability of rainfall, the statistical I-D rainfall thresholds may be also influenced by the locations of rain gauges, introducing uncertainties to any observation data (Nikolopoulos et al., 2014). So, it is technically challenging to


reliably define the statistical I-D threshold of debris flows for a specific gully or catchment. Most statistical I-D rainfall thresholds are only suitable for application to inform regional-scale early warning.

Moreover, the derivation of statistical I-D thresholds only focuses on the correlation between rainfall characteristics and debris flow occurrence. Although they are closely related to the initiation and occurrence of debris flows, the hydrological processes and land surface characteristics including topography and soil types are not considered when deriving statistical I-D models (Bogaard and Greco, 2018). It has already been demonstrated that statistical I-D thresholds may not always be

accurate in identifying the occurrence of debris flows and may lead to false detections (Berti et al., 2020). Such thresholds may only have regional validity in a catchment having mostly impervious terrain surface caused by a recent wildfire (Staley et al. 2017). It is suggested that approaches involving more input variables rather than just mean rainfall intensity and duration are needed to improve the accuracy of I-D thresholds (Hirschberg et al., 2021).

Related to rainfall-hydrological processes, the hydrodynamic forcing represented by unit-width discharge could be one of the important indicators controlling the occurrence of runoff-generated debris flows, which has been validated by laboratory experiments and in-situ observations (Tang et al., 2019; Tillery and Rengers 2019; Rengers et al., 2019; Wang et al., 2017; McGuire and Youberg, 2019; Gregoretti, 2000). This implies that a trigger-based threshold established by effectively taking

into account hydrodynamic conditions may be more reliable in predicting the occurrence of runoff-generated debris flows to support early warning. Attempts have been reported to analyze the initiation conditions for runoff-generated debris flows using a hydrological approach (Renger et al., 2016; Capra et al., 2018; Pastorello et al., 2020; Marino et al., 2022; Li et al., 2021; Bernard and Gregoretti, 2021). Hydrological and hydrodynamic models, e.g. HEC-HMS and FLO-2D, have been

used to predict the hydrological responses to rainfall in catchments prone to debris flows. However, direct measurement of flow discharge that is closely related to the initiation of runoff-generated debris flows in headwater catchments is still technically challenging, and high-quality monitoring data is rare. So, most of the hydrological or hydrodynamic models used in the previous studies were not properly calibrated and validated by field observations (Capra et al., 2018; Pastorello et al.,

2020), making the reliability of simulation results and the following analysis questionable. Attempting to overcome the problems, Gregoretti et al. (2016) built a weir at the outlet of a small headwater catchment to directly measure the flow discharge in a debris-flow source area.

In this study, we propose a new framework that integrates a hydrological and hydrodynamic model to estimate I-D rainfall thresholds for runoff-generated debris flows in a catchment. Unlike

the traditional statistical approaches that consider only meteorological factors, the proposed modelling framework effectively incorporates meteorological conditions, catchment topographic properties, and grain-size distribution of debris materials into the calculation of I-D rainfall thresholds, making it more suitable for application in areas with limited historical data.


## 2. The new framework

As illustrated in Fig. 1, the proposed framework aims to depict the rainfall-induced hydrological processes and estimate the I-D rainfall thresholds for runoff-generated debris flows by integrating hydrological and hydrodynamic predictions. The framework comprises four main components: rainfall estimation, hydrological analysis, hydrodynamic prediction, and quantification of hydrodynamic thresholds. Firstly, intensity-duration-frequency (IDF) analysis and a Gaussian

distribution profile are used to generate synthetic rainfall events. These synthetic rainfall events provide the meteorological inputs to the adopted hydrological model for predicting runoff in the debris-flow triggering areas. Driven by the discharge hydrographs of different return periods produced by the hydrological model as boundary conditions, a hydrodynamic model will be used to calculate the grid-based flow information including spatially and temporally varying flow depth and

velocity in the areas prone to debris flows. The produced flow information is then used to calculate the hydrodynamic metrics based on unit-width discharge for comparison with the corresponding hydrodynamic thresholds to indicate the occurrence of debris flows. In practice, it is not realistic to generate the I-D rainfall thresholds of debris flows at a cell scale. So, a zone threshold is further introduced to indicate the initiation of debris flows at a catchment scale. Combining the

hydrodynamic thresholds with the zone threshold, an integrated threshold is finally generated to predict the occurrence of debris flows.

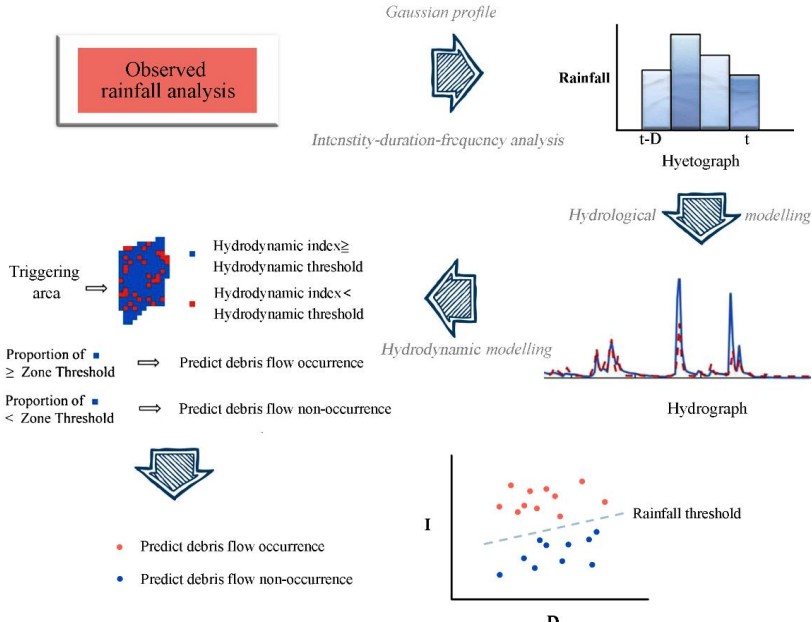

**Fig. 1 New framework based on an integrated hydrological and hydrodynamic modelling approach for estimating rainfall thresholds for the occurrence of runoff-generated debris flows.**



Compared with the previous similar studies which used peak discharge as the critical parameter for the prediction of debris flow occurrence (Li et al., 2020; Bernard and Gregoretti, 2021; Wei et al., 2018), the integrated hydrological and hydrodynamic modelling approach provides more reliable estimation by directly taking into account overland flow dynamics in the areas prone to debris flows. As the derived hydrodynamic indices are grid-based and the number of triggering cells can be
readily decided, the framework can be used to not only predict the occurrence but also evaluate the magnitude and scale of debris flows.

## 2.1 Rainfall analysis

The IDF (Intensity-Duration-Frequency) analysis plays a pivotal role in generating various synthetic rainfall events for driving hydrological modeling and subsequent analyses within the
proposed framework. The IDF curves employed in this study have been derived in accordance with the Atlas of Storms Statistical Parameters for Zhejiang Province, China, the region where our case study is situated (Zhejiang Province Bureau of Hydrology, 2003). This Atlas is a comprehensive compilation that draws from rainfall observations spanning the years 1953 to 2013 and serves as the authoritative reference for guiding hydraulic engineering design within Zhejiang Province.
Derived from the Pierson Type III probability density function, our IDF curves effectively encompass the range defined by the upper and lower bounds of curves generated by other types of probability functions, such as extreme value and long-term probability functions. As outlined in the Atlas of Storms Statistical Parameters, it is possible to compute extreme rainfall for various return periods and durations, denoted as $H_p$ as below:

$$H_p = K_p \bar{H} \qquad (1)$$

where $\bar{H}$ is the maximum average rainfall of a specific duration, $K_p$ is a coefficient with the subscript $p$ denoting the rainfall duration. The values of both $\bar{H}$ and $K_p$ can be obtained from the Atlas.

Nevertheless, it's important to note that the Atlas exclusively provides information on extreme rainfalls for durations of 1 hour, 6 hours, and 24 hours. Therefore, a method for estimating rainfall
with a 3-hour duration becomes imperative for the scope of this study. Within the Atlas, one can readily compute the rainfall depth ($H_i$) for rainfall events spanning durations ($t_i$) between 1 and 6 hours using the following approach:

$$H_i = H_{1h} t_i^{1-n_{1.6}} \qquad (2)$$

$$n_{1.6} = 1 + 1.285 \lg(H_{1h} / H_{6h}) \qquad (3)$$

where $H_{1h}$ and $H_{6h}$ are respectively the extreme rainfall depths of the 1h and 6h events, and $n_{1.6}$ is the corresponding attenuation coefficient.


Whilst the IDF analysis only specifies the total rainfall amount (i.e. depth) and duration, a Gaussian profile is further used to distribute the rainfall amount over a specific duration, following the approach used by Tang et al. (2019) and Berti et al. (2020). Compared with other similar studies which suggested the rainfall amount arbitrarily (McGuire and Youberg, 2019; Tang et al., 2019), the design hyetographs generated from IDF analysis may better reflect the actual rainfall characteristics
of the study area due to the use of local guidance created from multiple observations.

## 2.2 Hydrological analysis

The Nedbør Afstrømnings Model (NAM) (Madsen, 2000) is adopted to simulate the hydrological response to the design rainfall in the headwater catchment of the study site. The NAM model is part of the MIKE 11 river modelling system (Madsen, 2000), which was developed for
simulating rainfall-runoff process in sub-catchments and has been successfully applied in catchments across various climatic regimes including humid areas like the case study site (Butts et al., 2004; Nayak et al., 2013). The structure of the model mainly consists of four mutually interrelated storage components, i.e. snow storage (not used in this work), surface storage, lower zone (root zone) storage and groundwater storage, to account for different physical specifications in
the precipitation-runoff process (Makungo et al., 2010; Liu and Sun, 2010). The main inputs of the NAM model are rainfall and temperature (which is only needed when snow storage is considered and so it is not relevant in this work).

As NAM is a conceptual model, most of the model parameters are of empirical or conceptual nature and determined through calibration against hydrological observations. NAM model
calibration involves the optimization of multiple objectives that consider different aspects of a hydrograph: (1) water balance, (2) profile of the hydrograph, (3) peak flows, and (4) low flows. An automatic optimization procedure based on the shuffled complex evolution algorithm is introduced for solving the multi-objective calibration problem to support model calibration (Masen, 2000).

Table **1** lists the nine model parameters used in the simulations conducted in this work, which
are linked to the surface zone, the root zone, and the groundwater storage as mentioned.

**Table 1 Parameters of the NAM model involved in this work.**

| Parameter | Description | Limits of the parameters | |
| --- | --- | --- | --- |
| | | Lower bound | Upper bound |
| $U_{MAX}$ (mm) | Maximum water content in the surface storage. | 5 | 35 |
| $L_{MAX}$ (mm) | Maximum water content in the lower zone storage. | 50 | 350 |
| CQOF (–) | Overland flow runoff coefficient. | 0 | 1 |
| TOF (–) | Threshold value for overland | 0 | 0.9 |





| | | | |
|---|---|---|---|
| | flow. | | |
| TIF (–) | Threshold value for interflow. | 0 | 0.9 |
| TG (–) | Threshold value for recharge. | 0 | 0.9 |
| $CK_{IF}$ (h) | Time constant for interflow from the surface storage. | 500 | 1000 |
| $CK_{1,2}$ (h) | Time constant for overland flow and interflow routing. | 3 | 72 |
| $CK_{BF}$ (h) | Base flow time constant. | 500 | 5000 |

To evaluate the hydrological simulation results, two statistical indices are adopted, i.e. Nash-Sutcliffe Efficiency (NSE) coefficient (Nash and Sutcliffe 1970) and Schulz criterion (Schulz et al., 1999; Gregoretti et al., 2016). NSE has been widely adopted for evaluating the performance of hydrological models (Nayak et al., 2013; Makungo et al., 2010). The Schulz criterion has been used to validate hydrological simulations in small catchments prone to debris flows, similar to the current case study (Gregoretti et al., 2016). The Nash-Sutcliffe coefficient is defined as

$$NSE = 1 - \frac{\sum_{i=1}^{N}[q_o(i) - q_s(i)]^2}{\sum_{i=1}^{N}[q_o(i) - \overline{q}_o]^2} \qquad (4)$$

where $i$ is the data index; $N$ is the total number of data points; $q_o$ is the observed discharge (m³/s); $q_s$ is the simulated discharge (m³/s); and $\overline{q}_o$ is the average of the observed discharge (m³/s). The Schulz criterion is a model performance indicator defined as follows:

$$D = 200 \frac{\sum_{i=1}^{N}|q_s(i) - q_o(i)|q_o(i)}{N(q_{o,\max})^2} \qquad (5)$$

where $q_{o,max}$ is the observed maximum discharge (m³/s). The Schulz criterion classifies the performance of a hydrological model into four categories, ranging from very good to insufficient, as listed in Table 2.

**Table 2 Model performance classified by the Schulz criterion (Schulz et al., 1999)**

| Performance indicator | Very good | Good | Sufficient | Insufficient |
|---|---|---|---|---|
| D | 0-3 | 3-10 | 10-18 | >18 |

## 2.3 Hydrodynamic modeling

In the proposed modelling framework, a hydrodynamic model, the High-Performance Integrated hydrodynamic Modelling System (HiPIMS) (Xia et al. 2019), is employed to predict the grid-based flow information (i.e. water depth and velocity/discharge) in the debris flow triggering





area, driven by the output hydrograph(s) from hydrological modelling/analysis in the considered headwater catchment. HiPIMS solves the following fully 2D shallow water equations (SWEs):

$$\frac{\partial \mathbf{q}}{\partial t} + \frac{\partial \mathbf{f}}{\partial x} + \frac{\partial \mathbf{g}}{\partial y} = \mathbf{S}_b + \mathbf{S}_f \quad (6)$$

where $t$ is the time, $\mathbf{q}$ is the vector containing the flow variables, $\mathbf{f}$ and $\mathbf{g}$ are the flux vectors in the $x$ and $y$-directions, and $\mathbf{S}_b$ and $\mathbf{S}_f$ are the source term vectors representing bed slope and friction effect, respectively. The vector terms are given by

$$\mathbf{q} = \begin{bmatrix} h \\ uh \\ vh \end{bmatrix} \quad \mathbf{f} = \begin{bmatrix} uh \\ u^2h + \frac{1}{2}gh^2 \\ uvh \end{bmatrix} \quad \mathbf{g} = \begin{bmatrix} vh \\ uvh \\ v^2h + \frac{1}{2}gh^2 \end{bmatrix} \quad (7)$$

$$\mathbf{S}_b = \begin{bmatrix} 0 \\ -gh\frac{\partial b}{\partial x} \\ -gh\frac{\partial b}{\partial y} \end{bmatrix} \quad \mathbf{S}_f = \begin{bmatrix} 0 \\ -\frac{\tau_{bx}}{\rho} \\ -\frac{\tau_{by}}{\rho} \end{bmatrix} \quad (8)$$

where $h$ is the water depth, $u$ and $v$ are the two depth-averaged velocity components in the $x$ and $y$-directions, $\rho$ is the water density, $g$ is the gravitational acceleration, and $\tau_{bx}$ and $\tau_{by}$ are the frictional stresses estimated using the Manning equation:

$$\tau_{bx} = \rho C_f u \sqrt{u^2 + v^2} \quad \tau_{by} = \rho C_f v \sqrt{u^2 + v^2} \quad (9)$$

in which $C_f$ is the roughness coefficient calculated using

$$C_f = gn^2 / h^{1/3} \quad (10)$$


where $n$ is the Manning coefficient.

HiPIMS solves the above governing equations using a Godunov-type finite volume numerical scheme, making it suitable for simulating different types of shallow flow hydrodynamics, including the high-transient flash flooding processes induced by dam breaks or intense rainfall. HiPIMS is
also implemented on multiple graphics processing units (GPUs) to achieve high-performance computing and has been intensively tested for modelling catchment-scale overland flow and flooding processes as well as other types of flood hydrodynamics (Ming et al., 2022; Chen et al., 2022). HiPIMS is therefore suited for predicting the transient and complex flow hydrodynamics across different flow regimes in the debris flow triggering area, as required by this work. More
details of the model can be found in Xia et al. (2019) and Ming et al. (2020).



## 2.4 Hydrodynamic indices and thresholds

Previous studies have demonstrated that the transition from runoff to a debris-dominated flow may occur when the surface-water flow exceeds the thresholds of critical flow discharge (Gregoretti and Fontana, 2008; Gregoretti 2000; Recking 2009). Different formulae have been reported to estimate the critical discharge that triggers a runoff-generated debris flow. In this study, the equations proposed by Wang et al. (2017) and Whittaker and Jäggi. (1986) are considered. The formula introduced by Wang et al. (2017) calculates the critical unit-width discharge ($q_c$) as

$$q_c = 0.32 \frac{d_{84}^{2.5}}{(\tan \theta)^2 d_{16} C_u C_c^{0.4}} \quad (11)$$

where $\theta$ is the mean gradient angle of the triggering area, $d_{84}$ and $d_{16}$ are the 84% and 16% grain diameters in the particle size distribution curve and $C_u = d_{60}/d_{10}$ and $C_c = (d_{30})^2/(d_{60}d_{10})$ are the non-uniformity and curvature coefficients, with $d_{60}$, $d_{30}$ and $d_{10}$ respectively denoting the 60%, 30% and 10% grain diameters. Equation (11) explicitly takes into account inhomogeneity of sediment and has been shown to provide a reliable estimation of critical discharge (Wang et al. 2017). Most previous formulae are based on the mean grain diameter by assuming homogeneous or narrowly graded sands and therefore do not consider the effect of inhomogeneity of gully bed materials on debris flow initiation, which may potentially lead to less accurate results. Specifically relevant to the current study, the mean slope of the triggering area under consideration is within the range investigated by Wang et al. (2017).

The equation reported by Whittaker and Jäggi (1986) was developed to calculate the critical discharge that leads to the destabilization of artificial block ramps (hydraulic structures built with boulders to stabilize riverbeds), written as:

$$q_c = 0.257(s-1)^{0.5} g^{0.5} d_{65}^{1.5} (\tan \theta)^{-1.17} \quad (12)$$

where $s$ is the ratio of the sediment density ($\rho_s$) to water density ($\rho$), and $d_{65}$ is the 65% grain diameter in the particle size distribution curve. As Equation (12) was proposed to evaluate the erosion of block ramps with large blocks, it is used in this work to calculate the critical discharge that initiates the motion of large boulders in the triggering area.

In the implementation, the grid-based water depths and flow velocities predicted by HiPIMS are used to calculate the corresponding unit-width discharge $q$ (m²/s) at each grid cell in the triggering area as

$$q = h\sqrt{u^2 + v^2} \quad (13)$$

which is then used to define the hydrodynamic index in each grid cell and compared with the hydrodynamic thresholds (i.e. critical unit-width discharges) calculated using Equations (11) and (12) to indicate the potential occurrence of debris flows.





# 3. Case study

The proposed framework is applied to estimate the rainfall thresholds for triggering runoff-generated debris flows in a small catchment in Zhejiang Province, China.

## 3.1 Description of the study site

The study catchment is located in Fenghua City, Zhejiang Province, China. As shown in Fig. 2, the catchment has an area of about 0.17 km$^2$, and is crossed by a provincial road (No. 33 Provincial

Road) constructed in 2013. Downstream of the catchment, a countryside road connects the Linjiao Village to the outside. The area is dominated by a subtropical monsoon climate with most of the precipitation occurring in the summer months (1000–1700 mm of annual rainfall). In particular, the study catchment is often suffered from typhoons which may bring in extreme rainfall and cause flooding and other hydro-geohazards, e.g. debris flows. For example, the excessive rainfall

associated with Typhoon Fitow triggered a debris flow on 5th October 2013. The deposit fan of debris flow blocked the aforementioned countryside road, severely interrupting people's livelihoods. The increased risk of debris flows in the catchment is attributed to the availability of loose debris material disposed in channels (triggering area). The loose material was produced during the construction of Provincial Road No. 33, and can be easily eroded and transit into debris flows once

the necessary hydrodynamic conditions are met.

As clearly illustrated in Fig. 2, the study catchment is divided into two parts by the provincial road; south of the road is the headwater catchment area, and north of the road is the triggering area where the loose construction wastes are distributed. In Fig. 2, the top left image provides an aerial view of the study area, offering a comprehensive overview of the geographic context. The bottom

right image specifically focuses on the debris flow initiation area, providing a close-up view to highlight the key features and characteristics. When a large rainfall event hits the headwater catchment area, the induced overland flow will converge into the main channel. Through the culvert underneath the road, the flow will travel into the triggering area and erode the loose soil materials to create a large volume of water and sediment mixture, subsequently forming a debris flow. Table

3 summarizes the morphological characteristics of the catchment, extracted from the ASTER Global Digital Elevation Model (ASTER GDEM) of 5 × 5 m spatial resolution (Wei et al., 2018).

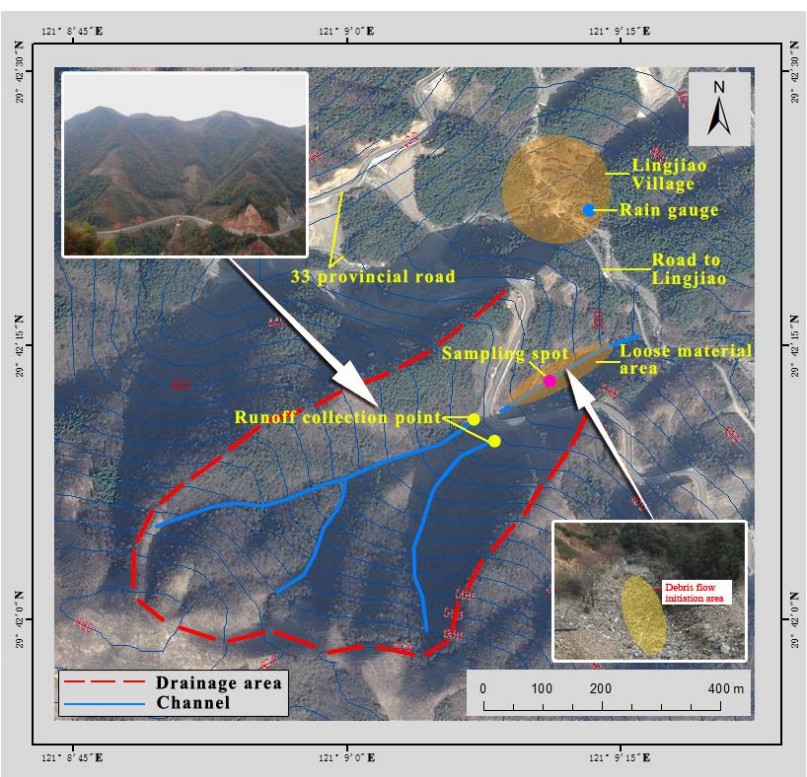

**Fig. 2 The study area and locations of monitoring instruments (modified from Wei et al., 2018).**

Grain-size distribution (GSD) has been demonstrated to be an effective index to characterize the rheological behavior of debris flows. Herein, we adopt a simple sieving method to obtain the GSD of the loose material. An approx. 0.1 m³ soil sample is taken from a 2 m × 1 m rectangular window at the site. The maximum grain size is analyzed to be about 120 mm whilst the minimum grain diameter is approx. 0.075 mm (Fig. 3). Bardou et al. (2003) classified debris flows into two main rheo-physical types, i.e. the viscoplastic class including the muddy debris flows that demonstrate a Herschel-Bulkley or Bingham flow behavior and the collisional class of stony debris flows that are featured with a Coulomb-like flow behavior. As shown in Fig. 3, the study area may be characterized as a collisional regime, most likely forming stony debris flows.

**Table 3 Morphological characteristics of the catchment ($A_c$ is the catchment area (km²); $\theta_{TRIG}$ is the average slope of the triggering area; $Y_{out}$ is the altitude at the outlet; $Y_{org}$ is the altitude at the channel head).**

| $A_c$ (km²) | $\theta_{TRIG}$ (°) | $Y_{out}$ (m) | $Y_{org}$ (m) |
|---|---|---|---|
| 0.17 | 18 | 384 | 668 |


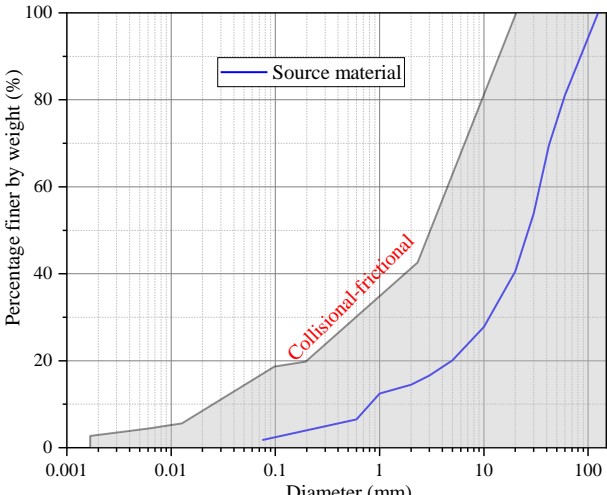

**Fig. 3 Grain-size distribution derived from the soil sample collected in the triggering area; the rheological classes proposed by Bardou et al. (2003) are added as a reference.**

## 3.2 Monitoring system

A monitoring system was set up to record rainfall and flow discharge in the case study site. For rainfall monitoring, a HOBO RGM-3 tipping bucket rain gauge produced by Onset Company, USA was installed. The rain gauge has a resolution of 0.2 mm per tip, meaning that the device will generate records for cumulative rainfall greater than 0.2 mm. Debris flows are usually triggered by locally convective rainfall that covers only a small storm cell (a few square kilometers or even less).

It is therefore important to install a rain gauge as close as possible to the triggering zone to ensure the reliability of rainfall records (Simoni et al., 2020). In this case, the rain gauge was installed on the roof of a house in Lingjiao Village (Fig. 2), which is only about 200 m away from the study site and can effectively avoid misrepresenting rainfall conditions.

        To measure flow discharge, two triangular weirs were installed at the outlets of the two

channels from the headwater catchment, just in front of the culvert (Fig. 4). The flow discharge is derived from a rating curve based on the measured water level. The rating curve used in this study is computed by solving the continuity equation recommended by Berti et al. (2020). To minimize water surface oscillations that may affect the reliability of water level measurement during extreme flow conditions, two approximately rectangular stilling basins (about 1.2 m wide and 2.0 m long)

were also constructed at the upstream sides of the weirs. The bottom of the stilling basins was flattened and built with concrete. The water level was measured using a pressure water-level recorder (Odyssey Capacitance Water Level Logger) installed inside a vertical PVC pipe, which has a resolution of approximately 0.8 mm. The PVC pipe was installed in each of the stilling basins to

improve stability, i.e. avoiding vibration of PVC pipe caused by the water flow. In addition, holes

were also created along the vertical direction to synchronize the changing water level with the

ambient flow (Wei et al., 2018). The data logger sampled the water level records at a 10-min interval.

The weirs were thin-walled with a 90° notch. The monitoring system operated successfully for about

one month, during which six rainfall events occurred, which provides short but reliable high-quality

measurements to support the current study.

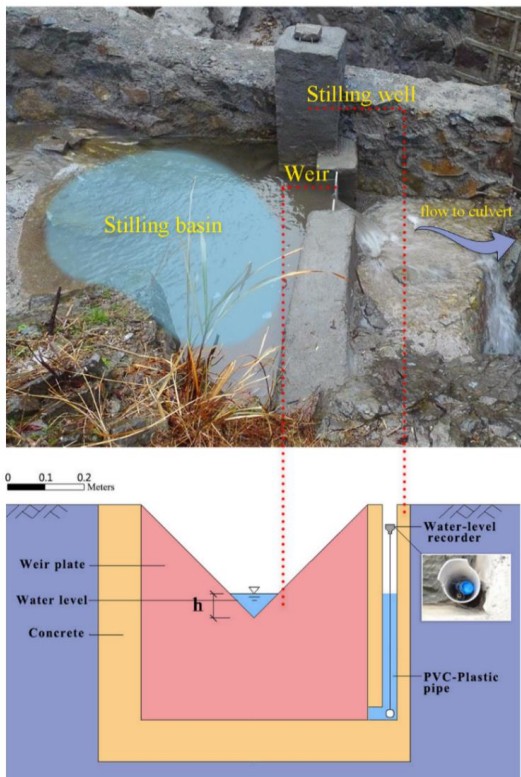


**Fig. 4 Water discharge measuring system (modified from Wei et al., 2018).**

The outflow discharge ($Q_W$) from the head catchment is estimated based on the continuity

equation:

$$Q_W(t) = \frac{dV_b}{dt} + Q_b \tag{14}$$

where $V_b$ is the volume of water in the stilling basin which is a function of the water depth, $dV_b/dt$

can be approximated using the backward finite difference method:

$$\frac{dV_b}{dt} \approx \frac{\Delta V_b}{\Delta t} = \frac{V_b(i) - V_b(i-1)}{10 \cdot 60} = \frac{A_b h_b(i) - A_b h_b(i-1)}{10 \cdot 60} \tag{15}$$





where $h_b$ is the water depth recorded at a 10-min interval, $i$ indexes the timestep, $\Delta t = 10 \times 60$ (s), $A_b = 2.4$ m$^2$ is the area of the stilling basin, and $Q_b$ (m$^3$/s) is the discharge over the weir calculated using the formula recommended by the Water Supply and Drainage Design Manual of China (Southwest Institute of Municipal Engineering Design and Research, 2000):

$$Q_b = 1.343(h/1000)^{2.47} \qquad (16)$$

The formula is applicable for thin-walled weirs with a weir angle of 90 degrees, and the water depth over the weir falls within the range of 0.02 m to 0.35 m. The discharge calculated using this equation should range from 8.9 x 10$^{-5}$ m$^3$/s to 0.1 m$^3$/s.

## 4 Results

In this section, the predicted flow discharge is first compared with the observation data to calibrate and verify the hydrological model. Then, rainfall events with and without triggering a debris flow are considered to validate the proposed modelling methodology. Finally, driven by the design rainfall events generated from IDF analysis, the proposed framework is applied to predict the rainfall thresholds for triggering debris flows. For the hydrological modelling, the time interval of the input rainfall data is 1h and the temporary resolution of the predicted hydrographs is 10 mins. The hydrodynamic modelling results are recorded every 10 min to maintain consistency with the hydrological modelling outputs.

### 4.1 Hydrological simulation results

The effective discharge measurement data is considered herein to evaluate the hydrological model introduced in Section 2.2, which covers six rainfall events over one month, as shown in Fig. 5. The monitoring period is divided into two parts, i.e. from July 3 to July 15 and from July 26 to August 3, to respectively use for model calibration and verification.

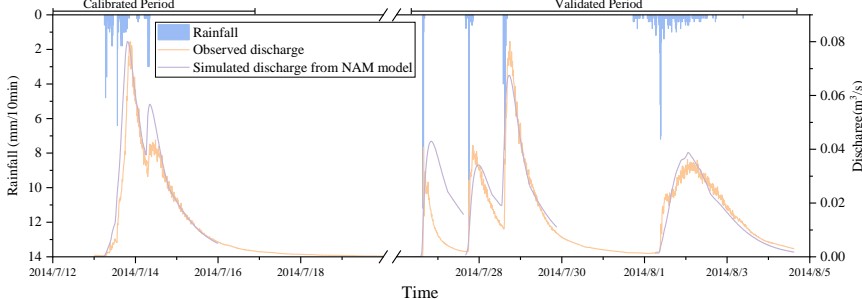

**Fig. 5 Recorded rainfall, observed discharge, and simulated discharge in the study site**

The NAM model is first calibrated automatically to decide the initial parameters, using an


automatic calibration scheme as introduced in Section 2.2. Subsequently, a trial-and-error method is further used to refine parameter values by visually comparing the simulated and observed hydrographs. Hydrological simulations in the study area were already conducted by Wei et al. (2018) using part of the hydrological monitoring data, which are enriched in this study using more monitoring data to further validate the hydrological model. The final values of the model parameters are listed in Table 4. Fig. 5 compares the predicted discharges with the observed data during the calibration process. The predicted flow discharges agree reasonably well with the observations. The model predicts a peak discharge at 0.081 $m^3$/s, which is close to the observed peak of 0.079 $m^3$/s. Quantitatively, the Nash-Sutcliffe coefficient ($NSE$) and Schulz criterion ($D$) calculated from the NAM predictions are presented in Table 5. In Table 5, the rainfall events on July 13 and July 14 of the calibration period and also the ones on July 27 and July 28 of the validation period are jointly assessed as they occurred close to each other and are reasonable to be considered as 'continuous' events. The NAM model returns 0.89 and 4.64 respectively for $NSE$ and $D$. According to the Schulz criterion, the performance of NAM model is ranked as 'good'.

**Table 4 Calibrated model parameters for the NAM model.**

| Parameter | Value |
|---|---|
| $U_{MAX}$ (mm) | 10 |
| $L_{MAX}$ (mm) | 100 |
| CQOF (–) | 0.96 |
| TOF (–) | 0.11 |
| TIF (–) | 0.21 |
| TG (–) | 0.66 |
| $CK_{IF}$ (h) | 754 |
| $CK_{1,2}$ (h) | 11.3 |
| $CK_{BF}$ (h) | 2441 |

**Table 5 The statistical matrices calculated from model calibration and verification.**

| Statistical indicators | Calibration period | | Verification period | |
|---|---|---|---|---|
| | July 13–15 | July 26–27 | July 27–29 | August 01–04 |
| NSE | 0.89 | -2.48 | 0.90 | 0.90 |
| D | 4.64 | 13 | 3.36 | 8.08 |

The calibrated NAM model is then applied to reproduce the 2$^{nd}$ part of the one-month measured data for model verification and the predicted discharges are compared with the observations. The discharge hydrograph predicted for 26$^{th}$ to 27$^{th}$ July does not compare well with the measurements, as reflected by the returned low value for NSE and high value for Schulz criterion, i.e. $NSE$ = -2.48 and $D$ = 13. The poor performance of the model for this specific event may be because the model is




not specifically calibrated for short-duration and high-intensity storms as the one under consideration (36 mm in 25 min). During the calibration period, the rainfall intensity was relatively low, and the runoff was mainly generated as a result of insufficient catchment storage following a sufficiently long rainfall event. However, for the event during 26-27 July, the intensity of the rainfall

was excessive and may be significantly greater than the catchment infiltration rate, subsequently generating excess infiltration runoff. Even so, the NAM model estimates the peak discharge to a reasonable level of accuracy, i.e. 0.041 m$^3$/s against the observed value of 0.043 m$^3$/s, and the relative error is only 5%.

From the results, as shown in Fig. 5, it can be seen that the NAM model performs better for the

rest of the verification period and satisfactorily reproduces the observed hydrographs, which is confirmed by the returned values of the statistical indicators, i.e. *NSE* = 0.90 and *D* = 3.3 and *NSE* = 0.90 and *D* = 8.1. Following model verification, it is recommended that the NAM model can effectively reflect the rainfall-runoff process in the study site and can be used in the following simulations and analysis.

Close examination of the numerical results can find that the NAM model may slightly overestimate or underestimate flood peaks in both the calibration and validation periods. The relative errors calculated against the observed and simulated peak discharges for the six rainfall events are 1.2%, 25%, 5%, 17%, 14% and 6%, respectively. Sensitivity analysis was previously conducted by Wei et al. (2018) to identify the key parameters influencing peak discharge calculation.

The results revealed that CQOF, $U_{max}$ and $CK_{12}$ exhibited a more profound influence on peak discharge calculation. Whilst the sensitivity analysis provided valuable insights, further research may be still needed in the future to investigate and confirm the performance of the model in reproducing catchment response to different rainfall patterns when more measured data becomes available to support model calibration and validation.

To further test the capability of the calibrated NAM model in simulating the hydrological response to short-duration and high-intensity rainfall as the 26-27 July rainstorm, an experiment is conducted to re-calibrate the model to the event. The comparison between observed and simulated discharge hydrographs for the event is shown in Fig. 6. It should be noted that only the 26-27 July rainstorm was used as the calibration process. It is clear that the NAM model performs better during

the re-calibration process and satisfactorily reproduces the observed hydrographs, which is confirmed by the returned values of statistical indicators, i.e. NSE = 0.59 and D = 10.4. The model parameters obtained after re-calibration are listed in Table 6. To further support model evaluation, the Kling-Gupta Efficiency (KGE') index is also considered for the re-calibration process (Kling et al., 2012). The KGE' values obtained in this study are around 0.56, providing an additional

assessment and confirmation of model performance.

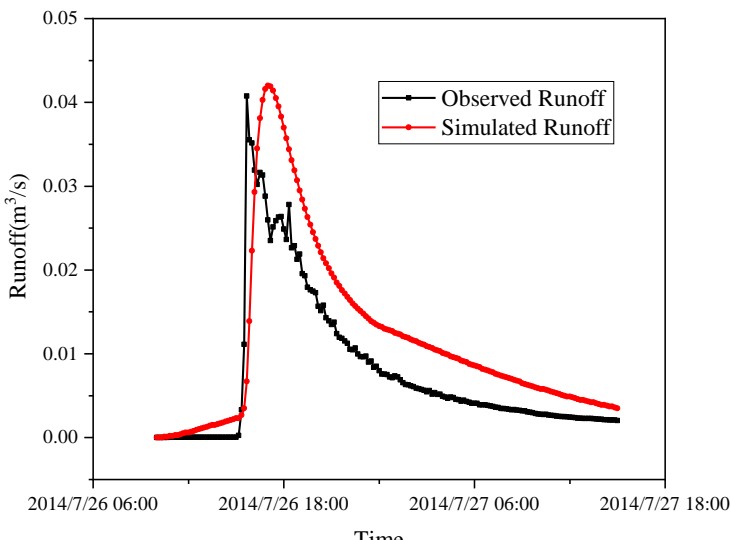

**Fig. 6 Comparison between observed and simulated hydrographs for the 26-27 July storm.**

**Table 6 The re-calibrated parameter values for the NAM model**

| Parameter | Value |
| --- | --- |
| $U_{MAX}$ (mm) | 10 |
| $L_{MAX}$ (mm) | 100 |
| CQOF (–) | 0.96 |
| TOF (–) | 0.11 |
| TIF (–) | 0.21 |
| TG (–) | 0.66 |
| $CK_{IF}$ (h) | 120 |
| $CK_{1,2}$ (h) | 5.2 |
| $CK_{BF}$ (h) | 2441 |


     From Table 6, it can be found that only the values of $CK_{1F}$ and $CK_{1,2}$ changed significantly whilst the other model parameters do not experience much change during re-calibration. The calibrated values of $CK_{1F}$ and $CK_{1,2}$ have changed from 754 and 11.3 respectively at the initial calibration to 125 and 5 following re-calibration. $CK_{IF}$ and $CK_{1,2}$ are the time constants related to

the routing of overland flow and may have a significant effect on the lag time between the timing of peak discharge and rainfall. The values of $CK_{1F}$ and $CK_{1,2}$ should therefore be carefully calibrated and checked for cases of short-duration and high-intensity storms. Based on the definition provided by Simoni et al. (2020), a rainfall burst or short-duration and high-intensity rainfall event occurs if the rainfall intensity reaches or exceeds 0.2 mm per 5 minutes (i.e. burst intensity threshold).

Following this definition, a rainfall event with a duration shorter than 1 hour and an intensity greater





than 25 mm/h may be classified as a short-duration, high-intensity rainfall event, e.g. the 26-27 July event considered in this work. The reproduction of the 26-27 July event indicates that the selected model is capable of simulating the hydrological responses to different hyetographs including short-duration, high-intensity rainfall events. Even though the occurrence frequency of short-duration and

high-intensity events is rare in the study area (See Supplement Fig. S1 and Fig. S2). The proposed framework may be used to estimate the initiation of runoff-generated debris flows under a wide range of rainfall conditions, including rainstorms leading to infiltration excess overland flows and those events causing saturation excess overland flows.

## 4.2 Validation of the proposed framework

In this section, the proposed methodology framework is tested for predicting a runoff-generated debris flow. As described in Equation 6, the initiation of runoff-generated debris flows is primarily influenced by the grain-size distribution and the slope of the channel. In this study, the initiation area is relatively small, so we did not take into account the spatial variations in the grain-size distribution. Instead, we treated the grain-size distribution as constant throughout the area. The slope

of the channel also plays a role in the initiation of debris flows. We analyzed the statistical features of the slope in the triggering area, using a digital elevation model (DEM) with a resolution of approximately 5m * 5m. The results indicated that the standard deviation of the slope was only about 3.2°. Therefore, we assumed that the slope can be adequately represented by its mean value, and we considered the hydrodynamic threshold to be spatially constant. Based on the grain size

distribution and the topography characteristics of the study area, the critical discharge calculated from Equation (11) is 0.024 $m^2$/s, which is defined as the hydrodynamic threshold $G$. To calculate the critical discharge mobilizing larger blocks, we follow a similar process as reported by Pastorello et al. (2020). The grain size is chosen to be 100 mm (i.e. $d_{65}$ = 100 mm as the representative size for boulders) as the maximum grain size in the triggering area is about 125 mm. Then, based on

Equation (12), the critical discharge for mobilizing sparse boulders is calculated to be 0.12 $m^2$/s, which is defined as the hydrodynamic threshold $W$.

In October 2013, a debris flow occurred in the case study site, triggered by the intense rainfall brought in by Typhoon Fitow. Unfortunately, no monitoring instrumentation was installed in the study area at the time, i.e. the catchment was ungauged. So, the rainfall data from the nearest station

(Huangtuling; 10 km away) is used. The characteristics of the rainfall event were analyzed by Wang et al. (2015), showing that the heavy rainfall was larger than 300 mm and covered an area of about 258 $km^2$ (including the current study area). The rainfall records from the Huangtuling station are considered to be relevant and used to drive the calibrated NAM model to predict the hydrograph out of the upper catchment, as shown in Fig. 7.


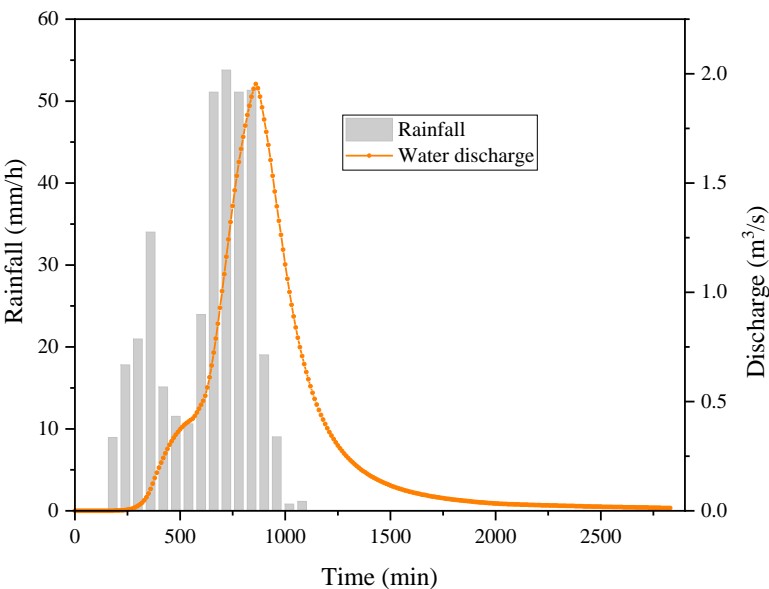


**Fig. 7 The rainfall input and predicted hydrograph during Typhoon Fitow in the study area.**

The predicted hydrograph is then used as the input to drive HiPIMS to predict grid-based hydrodynamic information (i.e. water depth and flow velocity) in the triggering area, discretized

using a DEM of 5 m spatial resolution. A uniform Manning coefficient of 0.04 is used to reflect the vegetation cover of the study area as suggested by Arcement and Verne (1989). From the output simulation results in terms of water depth and velocity, the unit-width discharges are calculated at each grid cell, across the entire simulation domain. Fig. 8 presents the distribution of the unit-width discharges from each cell in the triggering area at the time when the peak flow discharge is reached,

along with the threshold values.


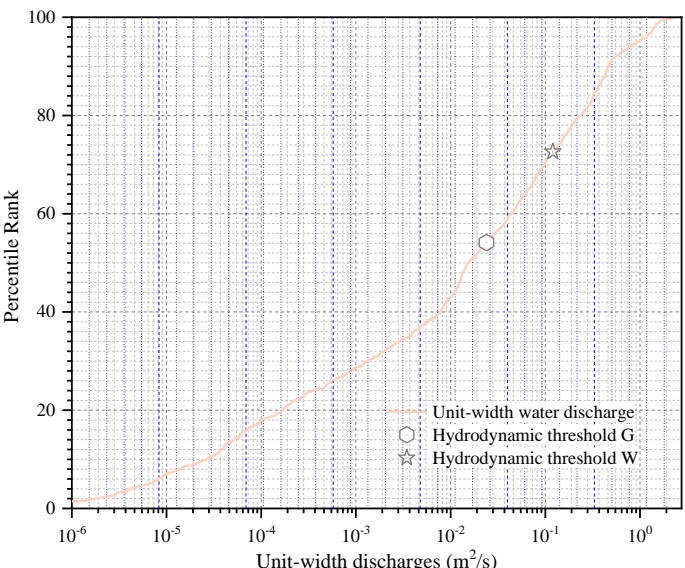

**Fig. 8 Distribution of unit-width discharges along with the threshold values.**

From Fig. 8, it can be seen that the unit-width flow discharge ranges from 0 to 2.42 m²/s across

the triggering area. From the discharge distribution curve, the corresponding percentiles of the unit-width discharge reaching the hydrodynamic thresholds *G* and *W* are 54% and 72%, respectively. This essentially means that 46% and 28% of the grid cells in the computational domain (i.e. triggering area) are predicted with a unit-width discharge larger than the hydrodynamic thresholds *G* and *W*. That is also to say that the hydrodynamic conditions for a runoff-generated debris flow

have been met in areas covered by 46% of the grid cells according to the hydrodynamic threshold *G*. Even based on the much higher threshold *W*, 28% of the grid cells have been predicted with the required hydrodynamic conditions. So, considerably large areas (at least 28%) are estimated to reach the required conditions that can trigger a debris flow. The results are consistent with the actual observation, i.e. a debris flow did occur during the typhoon event, demonstrating that the proposed

methodology can be used to predict the occurrence of runoff-generated debris flows.

We also consider the six rainfall events that did not trigger a debris flow to further test and confirm the predictability of the proposed framework. Fig. 9 shows the unit-width discharges predicted in each grid cell for the six events, along with the threshold values. The distributions of the unit-width discharges between the 13 July and 28 July events are very similar as the rainfall

peaks are almost the same. Table 7 further lists the relevant hydrological information. Among the six non-triggering rainfall events, it is observed that the lowest percentiles for the unit-width discharge reaching the hydrodynamic thresholds G and W are 95% and 99.3%, respectively. These percentiles indicate that only 5% and 0.7% of the grid cells inside the triggering area are predicted to reach or exceed the hydrodynamic thresholds of *G* and *W*. This implies that the hydrodynamic

conditions necessary for triggering a debris flow are met in only a small fraction of the grid cells





and the likelihood of debris flow occurrence is very low. This conclusion aligns with the actual observations, i.e. no debris flow was observed during these six rainfall events. These numerical tests demonstrate the framework's capability to accurately predict non-debris flow events and also confirm the reliability of the adopted hydrodynamic thresholds in indicating debris flow occurrence.

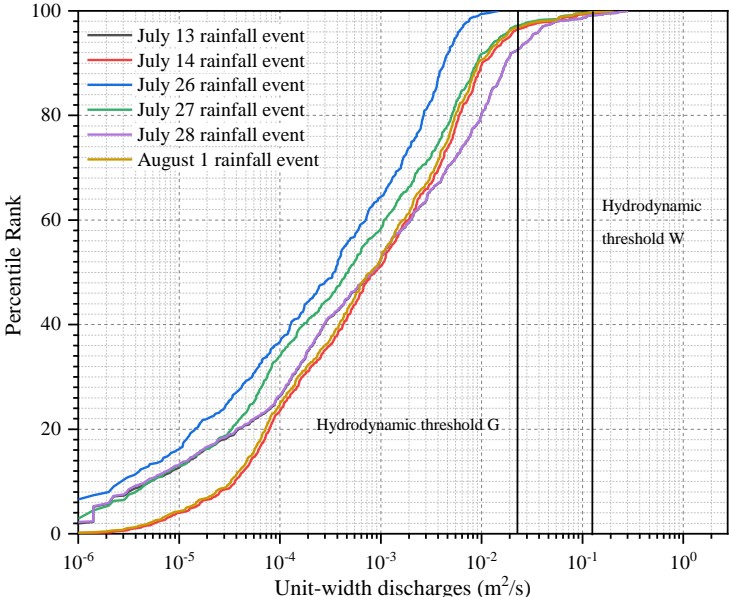


**Fig. 9 Distribution of unit-width discharges along with the threshold values predicted for the six rainfall events that did not trigger debris flow.**

**Table 7 Hydrological information of the six rainfall events that did not trigger a debris flow.**

| Rainfall event | Cumulative rainfall (mm) | Peak discharge ($m^3/s$) | Percentile of Threshold G (%) | Percentile of Threshold W (%) |
|---|---|---|---|---|
| July 13 | 51.4 | 0.079 | 95 | 99.3 |
| July 14 | 13 | 0.044 | 97 | 99.8 |
| July 26 | 34.8 | 0.040 | 100 | 100 |
| July 27 | 30.6 | 0.041 | 97 | 99.7 |
| July 28 | 27.6 | 0.080 | 95 | 99.3 |
| Aug 1 | 62.8 | 0.045 | 98 | 99.6 |


## 4.3 Estimation of rainfall thresholds using the proposed framework

Design rainfall of different return periods and durations is obtained for the study area through IDF analysis. In this study, we consider rainfall with a return period of 100, 20, 10, 5, 3 and 2 years





and a duration of 1, 3, 6, and 24 h. Therefore, a total of 24 design rainfall events are generated, as

illustrated in Fig. 10.

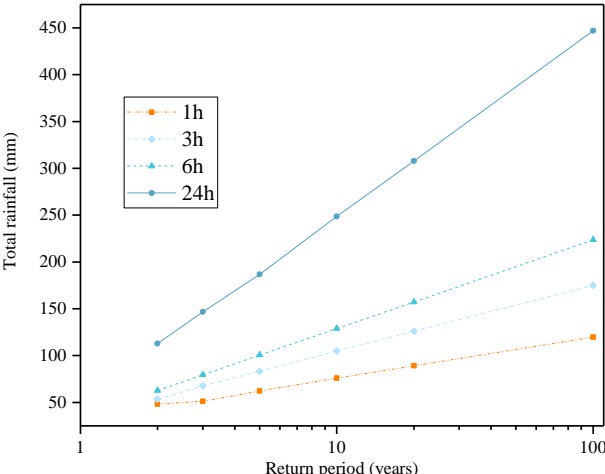

**Fig. 10 The design rainfall events obtained through IDF analysis.**

The 24 design rainfall events are then input to the NAM model to predict the corresponding flow hydrographs out from the headwater catchment, which are shown in Fig. 11. The resulting

rainfall profiles have different times to peak for design events of different rainfall durations. For example, the times to peak for the 3h rainfall and 6h rainfall are respectively 120 mins and 240 mins. From the results, it can be seen that shorter rainfall duration, e.g. 1 h or 3 h, leads to lower flow discharge, relative to an event with a longer duration (e.g. 6 h or 24 h). This is consistent with other studies in the literature. For example, Pastorello et al. (2020) reported a similar conclusion and

suggested that longer rainfall duration is needed to generate large enough flow discharge to mobilize large blocks.

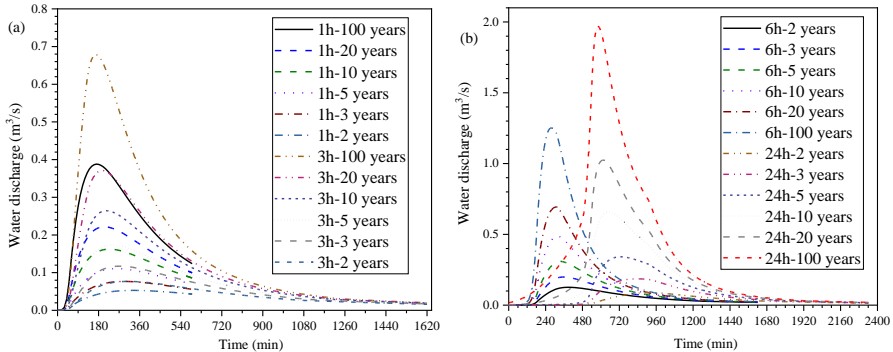

**Fig. 11 Predicted hydrographs for different design rainfall events: (a) 1h and 3h events; (b) 6h and 24h**

**events. In the legends, for example, "1h-100 years" means the rainfall event of 100 years return period and**

**1h duration.**

The predicted hydrographs are then used as the inputs for HiPIMS to predict the



corresponding grid-based flow information for calculating unit-width discharges. Fig. 12
shows the distributions of unit-width discharge for different design rainfall events along with
the relevant hydrodynamic thresholds. The results may be then analyzed to indicate the likely

occurrence of a debris flow.

The objective of this study is to calculate I-D rainfall thresholds to classify rainfall events
into two categories, trigger events and non-trigger events. Rather than predicting the
occurrence of debris flows at each grid cell, the focus is on determining whether a given rainfall
event will potentially trigger debris flows at the catchment scale. To achieve this, it is necessary

to establish a criterion based on a critical proportion of trigger cells in the triggering area to
determine whether a specific rainfall event can be classified as a trigger event for the entire
catchment. Following the approach reported by Zhao et al. (2020), such a critical proportion is
defined as the zone threshold for the triggering area, which can then be integrated with the
hydrodynamic thresholds to estimate a rainfall threshold.


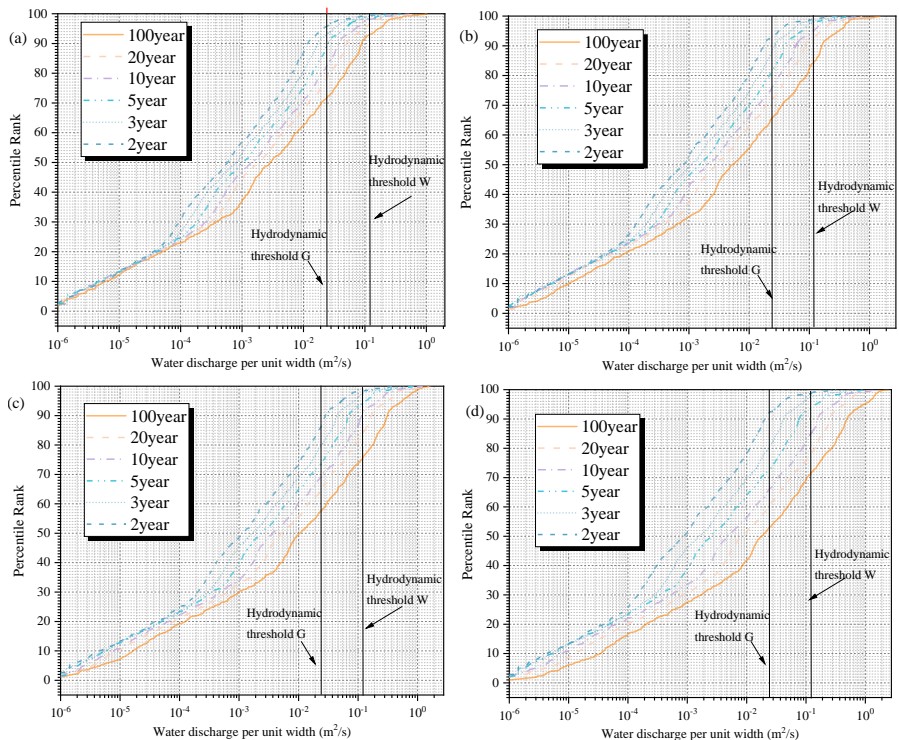

**Fig. 12 Distributions of unit-width discharge when the peak flow discharge occurs, along with**
**corresponding Hydrodynamic thresholds: (a) 1h rainfall events; (b) 3h rainfall events; (c) 6h rainfall events;**
**(d) 24h rainfall events.**

To define the zone threshold, the calculated grid-based unit-width discharges are


compared with the hydrodynamic thresholds G and W. If any of the thresholds are exceeded,
the associated grid cell is registered as a trigger cell. If the proportion of trigger cells in the
triggering area exceeds a critical value, i.e. zone threshold, a debris flow is considered to be
triggered; otherwise, it will be considered as a non-occurrence event (Fig. 1). In this study, a
proportion of 5% of the grid cells is used to define the non-triggering rainfall condition whilst
46% is used to define the triggering rainfall condition. Six different zone thresholds (i.e. 5%,
10%, 15%, 20%, 25%, and 30%) are tested to investigate their influence on the results. A trigger
rainfall event is identified if a zone threshold is exceeded (as presented in Fig. 12). In this way,
the rainfall thresholds associated with different zone thresholds can be decided, which are
shown in Fig. 13. The rainfall conditions of the Typhoon Fitow event is also included in Fig.
13, which has a return period of about 100 years.

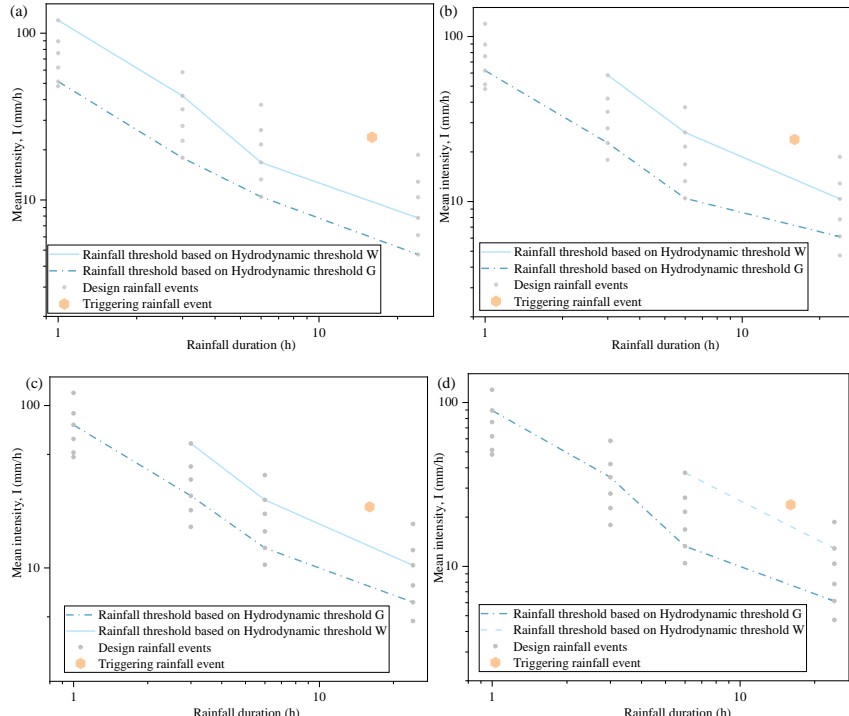



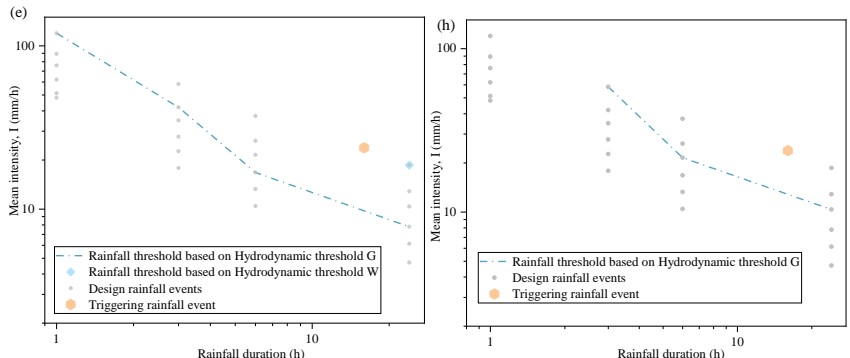

**Fig. 13 Rainfall thresholds associated with different zone thresholds: (a) 5%; (b) 10%; (c) 15%; (d) 20%; (e) 25%; (h) 30%.**

From Fig. 13, the calculated rainfall thresholds show an increasing trend as the zone threshold increases. This is as expected since a larger amount of rainfall is needed to generate more triggering cells. In addition, the rainfall conditions of Typhoon Fitow are beyond both the rainfall threshold based on threshold $G$ and threshold $W$ considering different zone thresholds. It means the proposed rainfall threshold is reasonable. The chosen values of zone threshold can also represent different levels of conservatism or adventurousness in the generated rainfall thresholds. A smaller zone threshold will lead to a lower rainfall threshold, indicating a more conservative approach. On the other hand, a larger zone threshold will create a higher rainfall threshold, which is less conservative. By associating different zone thresholds with the levels of warning, it will provide a framework for decision-making and response actions based on the identified rainfall thresholds.

Comparing the two adopted hydrodynamic thresholds, the rainfall thresholds calculated based on $W$ are normally larger whilst those calculated based on $G$ are smaller. The initiating mechanism to derive hydrodynamic threshold $G$ assumes progressive scouring occurs in sediment layers, which requires a lower critical discharge and subsequently a smaller amount of rainfall. Hydrodynamic threshold $W$ is built on the assumption of full bed failure, which needs a larger hydrodynamic force and a larger rainfall to trigger the failure. The intervals between the two corresponding rainfall thresholds are also related to the dynamics of a debris flow. At the beginning of a rainfall event, the resulting flow discharge is small, which increases as the rainfall amount and duration increase. When the discharge reaches the hydrodynamic threshold $G$, the first surge of debris flow may form although the volume is usually small. If the rainfall continues to intensify, the hydrodynamic conditions continue to evolve. When the maximum discharge exceeds the hydrodynamic threshold $W$, bed failure may occur, which will lead to a sudden increase in the debris flow volume. Mobilization of large blocks may worsen the situation and lead to the generation of the peak of the debris flow in terms of both flow and sediment volumes. Capra et al., (2018) investigated the temporal sequence of debris flows by





comparing monitoring data (including video images, seismic records and rainfall data) with the numerically predicted hydrologic response of the watershed under consideration. It was shown that the pulses of a debris flow event are not randomly distributed in time and the largest pulse is most commonly connected with the peak discharge.

Specifically, in Fig. 13(d), it can be seen that only three sets of design rainfall (24h-100 years, 24h-20 years and 6h-100 years) may trigger a debris flow associated with hydrodynamic thresholds $W$. When the zone threshold is taken to be 25%, only one design rainfall event (24h -100 years) can potentially trigger a debris flow. In Fig. 13(h), no design rainfall under consideration can trigger a debris flow again if the calculation is based on the hydrodynamic
threshold $W$.

    Herein, we also compare the proposed I-D rainfall thresholds with the regional rainfall intensity-duration (I-D) statistical thresholds. Fig. 14 presents the comparison between the proposed rainfall thresholds obtained with a zone threshold = 10% and the empirical I-D thresholds. The regional empirical rainfall thresholds were obtained after analyzing the main
characteristics (duration and intensity) of rainfall events that had triggered 1569 landslides (including many runoff-generated debris flows events) in Zhejiang Province during the period between 1990 and 2013 (Ma et al., 2015). Ma et al. (2015) also estimated the ID thresholds of 62 mountainous counties or cities in Zhejiang Province, including Fenghua City in which the study area is located. It should be noted that the I-D threshold from Ma et al. (2015) is the only
rainfall threshold developed in the study area. From Fig. 14, it is evident that both the proposed I-D rainfall threshold and the empirical rainfall threshold can effectively identify and distinguish triggering and non-triggering rainfall events, highlighting the reliability of the proposed framework. In addition, the present rainfall thresholds are located above the empirical thresholds when the rainfall duration is short (e.g. 1h and 3h). But as rainfall duration increases,
the empirical rainfall thresholds cross all three curves of the proposed rainfall thresholds and then they are located below the empirical thresholds. Using the proposed thresholds as references, the results indicate that the empirical thresholds may underestimate the occurrence of debris flow for short-duration rainfall events and overestimate the occurrence for longer-duration rainfall.

Wei et al., (2017) also calculated the rainfall threshold in the study area based on a runoff prediction model. The validated results demonstrated that both models, in this study and in Wei et al. (2017), were capable of predicting the occurrence of the real debris flow event. However, in Wei et al. (2017), the prediction of debris flow occurrence relied solely on the peak discharge in the channels. This approach allowed for identifying the presence of debris flow but did not
provide the means to assess its magnitude and scale. In contrast, this study employed an integrated hydrological and hydrodynamic modeling approach, which offers a more robust estimation by directly considering overland flow dynamics in areas susceptible to debris flows. Since the hydrodynamic indices derived from our model are grid-based and the number of triggering cells can be readily determined, our framework can not only predict the occurrence


but also evaluate the magnitude and scale of debris flows. As a result, the new proposed

framework offers a significant advantage in debris flow prediction compared to Wei et al.

(2017).

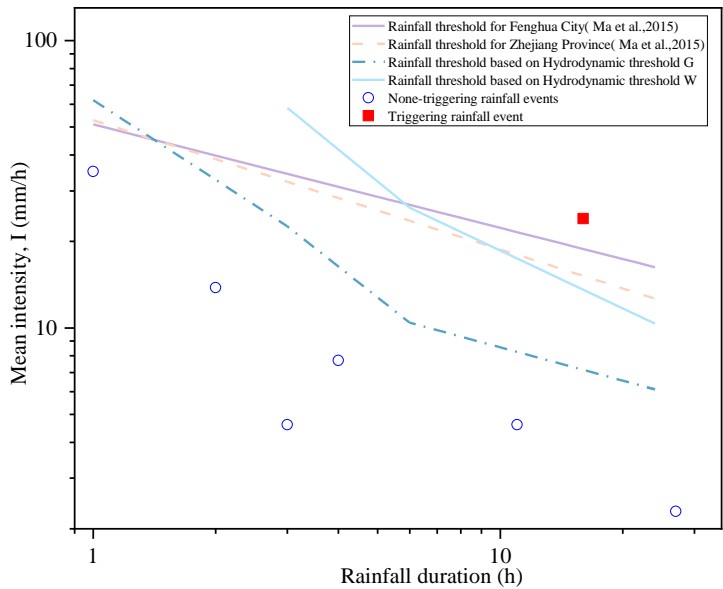

**Fig. 14 Comparison between present rainfall thresholds and the regional empirical ID thresholds**

## 5. Discussion


Model assumptions and physical processes in the targeted catchment will have a direct effect

on the final estimation of rainfall thresholds. For example, The grain size of sediment may increase

as surface flow washes away the fine particles. Such progressive coarsening of the debris flow

material is called "grain coarsening". Field observations suggested that such a process can be quick

and fine soil may be washed away over just a few years (Domènech et al., 2019). To evaluate the

potential uncertainties, the values of physical thresholds are changed from 50% to 200% with an

increment of 10%, creating a total of 16 physical thresholds. The estimated rainfall thresholds

corresponding to the different zone thresholds are shown in Fig. 15 and Fig. 16. For the

hydrodynamic threshold $G$, the values of critical threshold (CR) change from 0.012 m$^2$/s to 0.048

m$^2$/s; whilst for hydrodynamic threshold $W$, the CR varies from 0.06 m$^2$/s to 0.24 m$^2$/s.

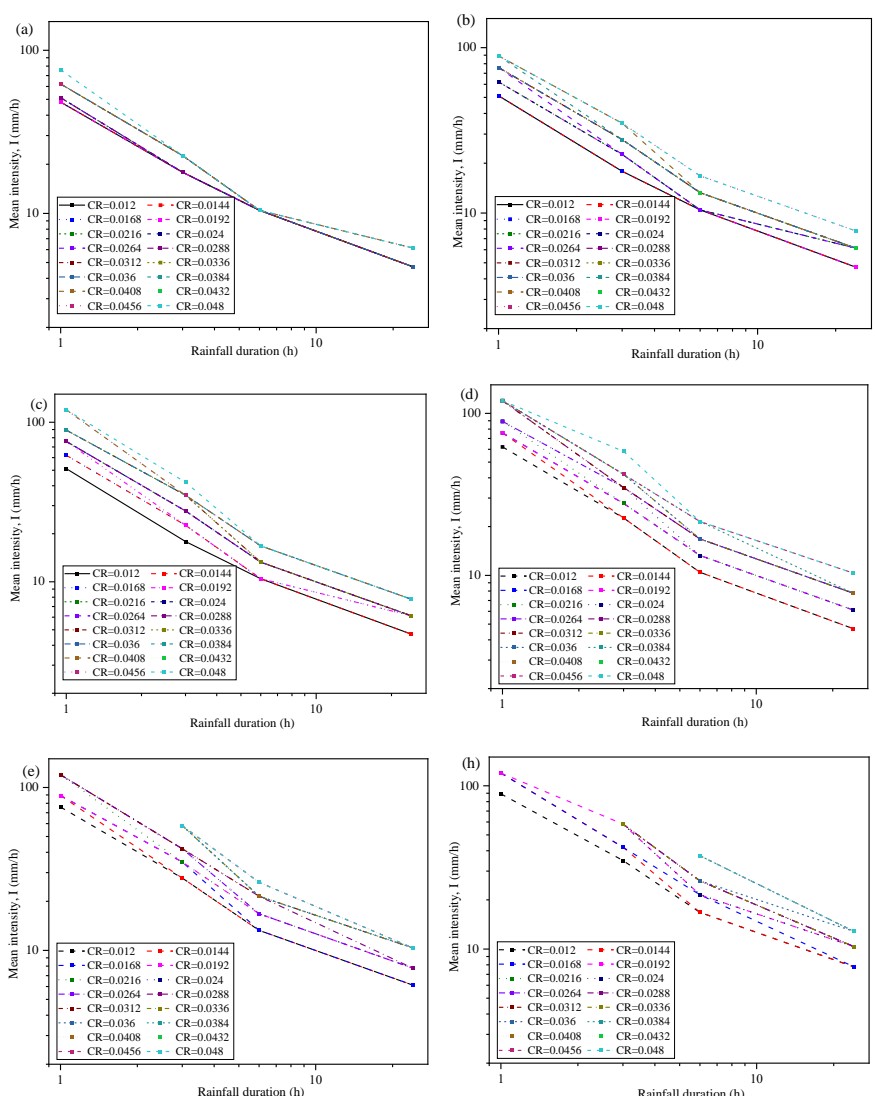

**Fig. 15 Sensitivity analysis of rainfall thresholds based on hydrodynamic threshold *G* associated with different zone thresholds: (a) 5%; (b) 10%; (c) 15%; (d) 20%; (e) 25%; (h) 30%.**

From Fig. 15, it is clear that the variation range of rainfall intensity is narrow when the zone threshold is small. When the zone threshold is 5%, the rainfall intensity only increases from 48 mm/h to 79 mm/h for 1h event; when the rainfall duration is 6h, there is not much change in the rainfall intensity even when the physical threshold changes significantly. Therefore, the rainfall intensity is not sensitive to the variation of the physical threshold when the zone threshold is small, indicating that it is not suitable to set the zone threshold to be overly small (like 5%). The variation range of rainfall intensity widens when the zone threshold increases. When the zone threshold is set to 20%, the rainfall intensity increases from 62 mm/h to 112 mm/h for 1h event and the change




becomes even greater for events with longer rainfall durations, e.g. 3h, 6h, and 24h. This indicates
that rainfall intensity is sensitive to the change of physical thresholds when the zone threshold is
sufficiently large.

Fig. 16 shows the variation of the rainfall intensity following the change of hydrodynamic
threshold $W$. Compared with the results related to hydrodynamic threshold $G$ as shown in Fig. 15,
the rainfall intensity is found to vary in a wider range even when the zone threshold is small (Fig.
16a). When further increasing the zone threshold, fewer rainfall events can trigger debris flows.
When the zone threshold reaches 30% (Fig. 16h), no rainfall event can induce debris flows when
the critical threshold is assumed to be larger than 0.096 $\text{m}^2/\text{s}$.

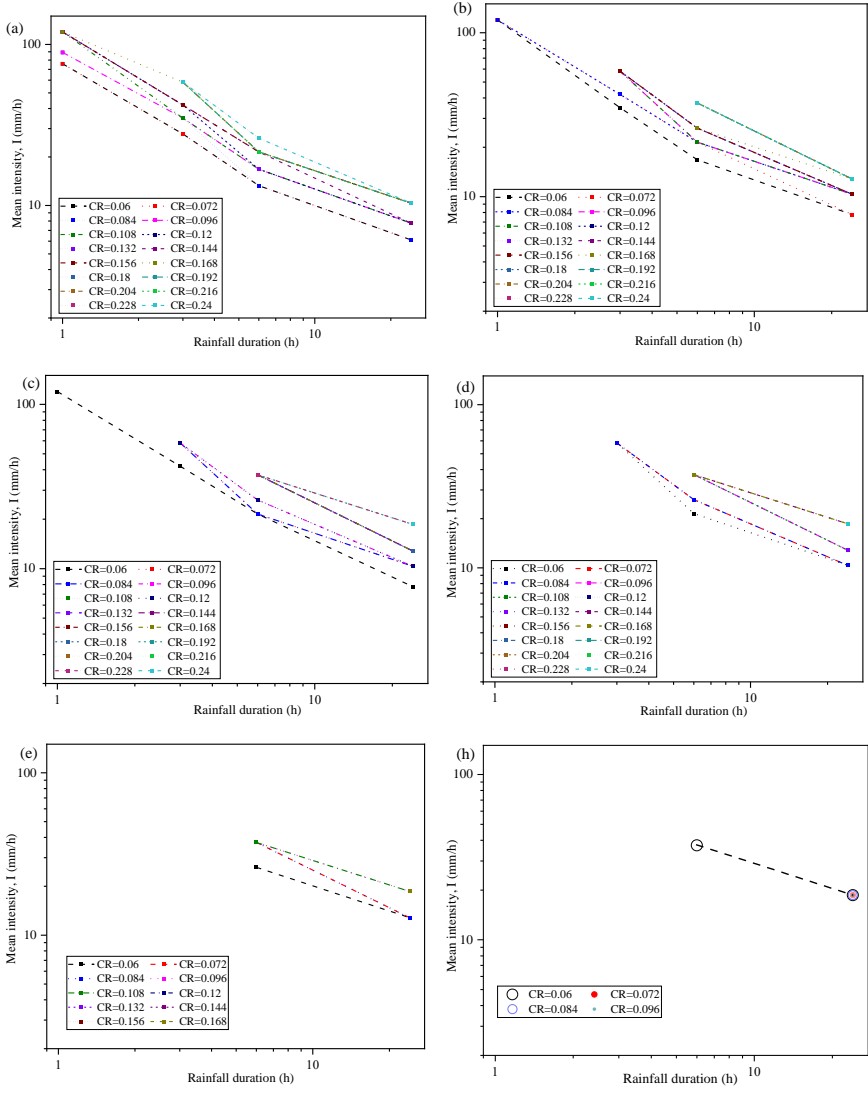




**Fig. 16 Sensitivity analysis of rainfall thresholds based on hydrodynamic threshold $W$ associated with different zone thresholds: (a) 5%; (b) 10%; (c) 15%; (d) 20%; (e) 25%; (h) 30%.**

The importance of hydrological and hydrodynamic processes in triggering runoff-generated debris flows has been recognized and discussed. However, due to the scarcity of observed flow data but better availability of rainfall data, the statistical I-D rainfall thresholds are still mostly used although their development does not consider the hydrological process and flow hydrodynamics. Clear limitations have been noticed in the use of statistical I-D rainfall thresholds to predict the occurrence of runoff-generated debris flows. Firstly, in most

of the existing studies, subjective approaches are used to identify the rainfall events and rainfall conditions that trigger debris flows. The lack of objective criteria may bring uncertainties to the definition of rainfall thresholds, limiting their applicability in early warning systems. Secondly, the reliability of statistical I-D rainfall thresholds depends entirely on the datasets being used and high-quality long-term observations are essential to derive reliable thresholds.

The approach therefore cannot be applied to ungauged catchments where historical data is missing.

For the initiation of landslide-triggered debris flows or shallow landslides, infiltration plays a crucial role (Fan et al., 2020). Therefore, accurately simulating the infiltration process in unsaturated soils is vital for establishing the rainfall threshold for landslides-triggered debris flows or shallow

landslides. On the other hand, the most significant triggering factor for runoff-generated debris flows is the surface flow dynamics. The transition from a clear water flow to a debris flow may occur when the hydrodynamic conditions of the flow (e.g. flow discharge) exceed certain thresholds. Therefore, spatially distributed hydrodynamic information (e.g. water depth and velocity) can provide important information for estimating the occurrence of runoff-generated debris flows. The

hydrodynamic information may be produced using a hydrological or hydrodynamic approach and used to estimate rainfall thresholds indicating debris flow occurrence.

In this work, a new approach is explored to predict the potential occurrence of runoff-generated debris flows by integrating hydrological and hydrodynamic models to predict rainfall-induced hydrological response and the resulting surface flow hydrodynamics. Compared with the traditional

statistical I-D analysis approaches that only consider meteorological factors, the proposed modeling framework can effectively take into account the meteorological conditions, topographic properties of the targeted catchment, and grain-size distribution of debris materials. The use of a fully physically based hydrodynamic model enables the proposed framework to generate rainfall thresholds in areas with limited historical data on debris flow occurrence. As the hydrodynamic

thresholds (e.g. critical discharge) should not vary against the hydrological properties of the catchments, the framework can be readily applied to other similar catchments (e.g. Alpine region) when essential data are available for model calibration and setup. However, it should be noted that the proposed approach is more suited for the cases where the initiation area and headwater catchment area are easy to identify. Such debris flows are different from those initiated "by rilling". Rills are


common in steep slopes and usually form complex and highly connected distributary networks. Although debris flows in such catchments may still be triggered by overland flows, they involve a gradual transition from clear water flow to debris flow, and it is therefore difficult to identify the source and triggering areas (Berti et al., 2020). In such cases, hydrological analysis is far more complicated and extensive field surveys are necessary to identify the initiating areas.

Although the proposed approach can entail the hydrological processes related to the initiation of runoff-generated debris flows and better represent the underlying physics, there are still some limitations. The main limitation is that the proposed framework is only applied and tested in one case study catchment due to the challenge of collecting high-quality observed hydrological data and debris-flow data in small and unstable channels. If possible, the proposed framework should be

tested in other catchments with more hydrological and debris-flow observations to further confirm its robustness. The adoption of a conceptual NAM hydrological model represents another limitation of the proposed modelling framework. The model adopts the bucket-style description of the hydrological computation units (HCUs), where the catchments or sub-catchments are treated as HCUs, overlooking essentially physical characteristics inside these units. Ideally, a physically-based

hydrological model should be used. However, a physically-based model possesses many parameters that represent the physical characteristics of the catchment and need to be determined through field measurements. The lack of abundant field measurements imposes challenges in properly calibrating a physically based model in this work. Consequently, all of the model parameters can only be determined solely through simple calibration and the adopted NAM represents a suitable choice for

the current application. This work aims to propose a new framework for estimating rainfall thresholds for debris flow occurrence. The NAM model can be replaced by a physically based model to simulate hydrological response in the future if essential observation data becomes available. Actually, attempts have been made to directly apply hydrodynamic models to simulate the whole flooding process from rainfall runoff, overland flow to inundation in data-rich catchments (Ming et

al. 2020). It is expected that data scarcity will become less of an issue in the future with the increasing availability of high-resolution remote sensing data, e.g. LiDAR data.

## 6. Conclusions

   The occurrence of runoff-generated debris flows is recognized to be closely related to the surface flow hydrodynamics following a rainfall event. This work presents a novel framework

to estimate the rainfall thresholds that trigger runoff-generated debris flows by comparing hydrodynamic indices with threshold values. An integrated hydrological and hydrodynamic modelling approach is used to calculate the grid-based hydrodynamic indices, i.e. unit-width discharges, in the triggering area, which are compared with the specific hydrodynamic thresholds to predict the occurrence of debris flows. The integrated modelling framework can

reliably predict the spatio-temporal varying hydrological process and hydrodynamics driven



by meteorological inputs and influenced by topographic properties of the catchment, which is superior to the traditional statistical ID thresholds derived from rainfall conditions without considering these factors.

The proposed approach has been validated and applied to derive rainfall thresholds for runoff-generated debris flows in a small catchment in Zhejiang Province, China. Due to the use of a physically based hydrodynamic model to predict the rainfall-induced flow hydrodynamics, the approach may be used to estimate rainfall thresholds in areas where there is a lack of observational records of debris flow occurrence.

## Acknowledgements

This research is financially supported by the National Natural Science Foundation of China (Grant No. 42230702 and 42307262 ), UK Natural Environment Research Council through the SHEAR project WeACT (NE/S005919/1), State Key Laboratory of Geohazard Prevention and Geoenvironment Protection Independent Research Project (No. SKLGP2021Z024) and the Natural Science Foundation of Sichuan Province (Grant No. 2022NSFSC1129).

## Author contributions


All authors contributed to the study conception and design. Material preparation, data collection and analysis were performed by Zhen-lei Wei, Yue-quan Shang and Xilin Xia. The first draft of the manuscript was written by Zhen-lei Wei and Qiuhua Liang. All authors commented on previous versions of the manuscript. All authors read and approved the final manuscript.

## Declaration of interests


The authors declare that they have no known competing financial interests or personal relationships that could have appeared to influence the work reported in this paper.

## Data availability

All data can be provided by the corresponding authors upon request

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
