# Peer review of "A coupled hydrological and hydrodynamic modelling approach for estimating rainfall thresholds of debris-flow occurrence"

_Natural Hazards and Earth System Sciences, 2023_

## Author Response (AR1)

**Revision notes on** Manuscript No. NHESS-2023-180

First of all, the authors thank the Editors and the reviewers for considering our manuscript and providing constructive comments to help us improve the quality of our work. We have accordingly revised the manuscript by carefully addressing or answering the comments point-by-point, summarized as follows. Following the revision, we hope we have clarified all of the points summarized by the Editor and reviewers.

**Responses to the Comments Raised by the Editor**

Your manuscript underwent a thorough review by two experts, both of whom raised concerns about the novelty of your work. Upon examining your responses, the manuscripts, and your previous studies, it appears that you have assembled a modeling chain that integrates components previously addressed in some of your earlier papers.

1. While you have discussed the novelty arising from this approach, it seems necessary to enhance clarity in presenting the uniqueness of your analysis, particularly in the introduction and throughout the manuscript. Consequently, I would like to offer you the opportunity to submit a substantially revised version, addressing the comments provided by the referees and placing a stronger emphasis on explaining the novelty of your work.

**Authors' reply:**

Thanks for the comments. We have added a detailed and stronger emphasis on explaining the novelty of our work in the Introduction, and Discussion part as below:

**In the Introduction part:**

*"Peak discharge has garnered widespread acceptance as a standard critical parameter for predicting debris flow occurrences (Wei et al., 2018). For instance, Li et al. (2021) established rainfall intensity-duration thresholds based on process-based critical runoff discharge. Bernard and Gregoretti (2021) proposed an approach to determine debris flow occurrence through coupling a hydrological model with a critical discharge relationship using rainfall and raw radar data. However, in these existing frameworks,*

*the peak discharge is usually predicted by a hydrological model; such an approach may predict the occurrence but not the scale of debris flow".*

*"Moreover, in comparison with the previous studies that solely used peak discharge as a critical parameter for predicting debris flow occurrence (Li et al., 2020; Bernard and Gregoretti, 2021; Wei et al., 2018), the current integrated hydrological and hydrodynamic modeling approach offers potentially a more detailed and reliable estimation by directly considering overland flow dynamics in susceptible debris flow areas. With grid-based hydrodynamic indices and through identifying the spatial distribution of triggering cells, the proposed framework facilitates the prediction of occurrence, and meanwhile, the magnitude and scale of debris flows".*

**In the Discussion part:**

*"Actually, several studies have been reported to establish Intensity-Duration (ID) rainfall thresholds through a numerical approach (Domènech et al., 2019). In the previous studies, runoff-induced erosion is considered to occur when the bed shear stress exceeds a critical value, and the volumetric concentration of solids in the debris flow is smaller than an equilibrium value. Furthermore, most of the previous studies adopt simplified hydrological simulations, e.g. calculating runoff using a basic lumped infiltration model that neglects the initial moisture content of the soil. Different from these existing attempts, the proposed approach focuses on predicting spatially varying hydrodynamic index (unit-width discharge) at each cell to indicate the occurrence of runoff-generated debris flows.*

*In the authors' previous works (Wei et al. 2018; Wei et al. 2017), the rainfall thresholds in the same study site were also calculated using a runoff prediction model. Wei et al. (2018) developed an approach solely based on a hydrological model, whilst Wei et al. (2017) presented a machine learning model for runoff prediction. These approaches are clearly different from the current integrated hydrological and hydrodynamic modeling framework which provides a more robust method to directly incorporate overland flow dynamics into debris flow occurrence estimation. Furthermore, our previous studies used peak discharge as the critical parameter to indicate debris flow occurrence; if the peak discharge predicted by the adopted hydrological model exceeds*

*the critical discharge, debris flow occurrence is confirmed. Whilst such peak discharge-based approaches can estimate debris flow occurrence, they cannot provide any insights related to the magnitude and scale of a debris flow. Our new framework includes the use of a hydrodynamic model to predict detailed overland flow dynamics to derive grid-based hydrodynamic indices in the areas susceptible to debris flows. This enables not only the prediction of debris flow occurrences but also provides insights into the magnitude and scale of a debris flow.*

*"Furthermore, to evaluate debris flow occurrence at a catchment scale, we introduce a new concept as "zone threshold" to represent varying degree of conservatism or adventurousness in rainfall thresholds. By associating different zone thresholds with the corresponding level of warning, the framework facilitates decision-making and response actions based on identified rainfall thresholds, allowing implementation of risk management strategies tailored to the different level of caution or preparedness."*

The Editor also could find the changes in Lines 93-99, Lines 105-113 and Lines 697-725 of the revised MS.

2. Furthermore, it would be beneficial to provide additional clarification regarding the hydrodynamic model's input. Specifically, could you elaborate on whether the hydrograph input is distributed, or if you have selected specific input points along the river network or elsewhere?

**Authors' reply:**

Thanks for the comments. The input hydrograph is applied at a particular input point/cell located at the outlet of the culvert beneath the road as the point-source boundary conditions to drive HiPIMS to simulate the subsequent flow dynamics. During an intense rainfall event that occurs in the headwater catchment area, the generated overland flow converges into the main channel, passes through the culvert beneath the road, reaches the triggering area, and erodes the available loose soil materials to initiate a debris flow.

The Editor also could find the changes in Lines 483-488 of the revised MS.

**Responses to the Comments Raised by Reviewer #1**

The authors present a method to obtain rainfall intensity-duration thresholds for runoff-generated debris flows. To obtain these thresholds the authors have used existing equations in the literature to compute the critical discharge that could lead to the destabilization of catchment slopes that have a fine granulometry and slopes that have large boulders. The presented rainfall intensity-duration thresholds have been established using the rainfall events that reach the critical discharge value over different percentages of pixels in the catchment. However, no recommendation on the critical area required for debris flow initiation is given.

1. My main concern is the novelty of the presented work. The hydrological simulations were conducted by Wei et al., (2018). The relationship between the intensity-duration of the rainfall event that triggered the 2013 debris flow and the discharge is also studied in Wei et al., (2018). Moreover, as the authors point out in line 630, Wei et al., (2017) already used the same method to establish an intensity-duration threshold for the studied catchment.

**Authors' reply:**

Thank you for your comments. The authors have explained the novelty and advantages of the newly proposed model in lines 630-645 of the initial manuscript. The framework presented in this study significantly diverges from our previous research. Firstly, in earlier studies led by Wei et al. (2018), the focus was solely on the hydrological model (NAM model), with the presentation of hydrological simulations. In Wei et al. (2017), a machine learning model was employed for runoff prediction. In contrast, our current study adopts an integrated hydrological and hydrodynamic modeling approach, providing a more robust estimation by directly incorporating overland flow dynamics in regions prone to debris flows.

The second key deviation lies in our approach to determining the occurrence of debris flow. In prior studies, such as Wei et al. (2018), peak discharge was used as the critical parameter. If the calculated peak discharge from the hydrological model exceeded the critical discharge, debris flow occurrence is confirmed; otherwise, it was deemed non-occurrence. However, in this study, we use hydrodynamic metrics as the critical

parameter for predicting debris flow occurrences. This is in contrast to the common practice, such as our earlier works (Wei et al., 2018 and Wei et al., 2017), where peak discharge served as the critical parameter. Now peak discharge has become widely accepted as the standard critical parameter for predicting debris flow occurrences. For instance, Li et al. (2021) established rainfall intensity-duration thresholds based on process-based critical runoff discharge. Bernard and Gregoretti (2021) proposed determining debris flow occurrences through a hydrological model coupled with a critical discharge relationship using rainfall and raw radar data. However, it is important to note the limitations of the previous frameworks, which solely relied on peak discharge predicted only by a hydrological model. Such an approach could predict the occurrence of debris flow but lacked the capability to predict the scales of debris flow. In essence, our previous model could only forecast the likelihood of debris flow occurrence, without providing insights into the magnitude of the runoff-generated debris flow.

In this study, we introduce a novel framework that employs an integrated hydrological and hydrodynamic modeling approach to enhance the accuracy of rainfall thresholds estimation for runoff-generated debris flows. The incorporation of a hydrodynamic model enables the prediction of detailed flow dynamics, providing grid-based information such as water depth and flow velocity in regions susceptible to debris flows. The flow information obtained is then employed to compute hydrodynamic metrics based on unit-width discharge. These metrics are compared with corresponding hydrodynamic thresholds, serving as indicators for the occurrence of runoff-generated debris flows. Given that the derived hydrodynamic indices are grid-based, the framework allows for determining the proportion of trigger cells within the triggering area or the total number of trigger cells. This leads to clear advantages of the current framework against existing approaches, enabling not only the prediction of debris flow occurrences but also providing insights into the magnitude and scales of debris flow. As a summary, in contrast to our previous study and similar works, such as Li et al. (2021), our approach represents a significant advancement in the prediction of runoff-generated debris flows.

Furthermore, to evaluate debris flow occurrences at the catchment scale, we have introduced a new concept as "zone threshold." The zone threshold is defined as the critical proportion of trigger cells within the triggering area. If the proportion of trigger cells surpasses the zone threshold, it signifies a debris flow occurrence; otherwise, it is categorized as a non-occurrence event. This concept integrates zone thresholds with hydrodynamic thresholds within the proposed framework. The advantage of incorporating zone thresholds lies in their ability to represent varying degrees of conservatism or adventurousness in rainfall thresholds. A smaller zone threshold corresponds to a lower rainfall threshold, reflecting a more conservative approach. Conversely, a larger value results in a higher rainfall threshold, indicating a more adventurous approach. By associating different zone thresholds with corresponding levels of warning, the framework facilitates decision-making and response actions based on identified rainfall thresholds. This approach allows for a spectrum of risk management strategies tailored to the desired level of caution or preparedness.

Overall, with the integration of the hydrodynamic model, the introduction of a critical parameter, and the incorporation of zone thresholds, we assert that our study makes a novel contribution to the prediction of runoff-generated debris flows.

The Reviewer also could find the changes in Lines 93-99, Lines 105-113 and Lines 697-725 of the revised MS.

*References:*

*Wei, Z.L., Shang, Y.Q., Zhao, Y., Pan, P. and Jiang, Y.J., 2017. Rainfall threshold for initiation of channelized debris flows in a small catchment based on in-site measurement. Engineering Geology, 217:23-34.*

*Wei, Z.L., Xu, Y.P., Sun, H.Y., Xie, W., Wu, G., 2018. Predicting the occurrence of channelized debris flow by a cascading flood debris-flow model in a small debris flow-prone catchment. Geomorphology, 308 :78-90*

*Li, Y.J., Meng, X.M., Guo, P., Dijkstra, T., Zhao, Y., Chen, G., Yue, D.X., 2021. Constructing rainfall thresholds for debris flow initiation based on critical discharge and S-hydrograph. Engineering Geology, 280:105962.*

*Bernard, M. and Gregoretti, C. 2021. The use of rain gauge measurements and radar data for the model-based prediction of runoff-generated debris-flow occurrence in early warning systems. Water Resources Research, 57(3): e2020WR027893.*

2. An additional major concern is that the thresholds presented in this manuscript have only been tested using one debris flow event. However, the authors employ rather strong language throughout the manuscript and claim that the proposed thresholds can effectively identify the triggering and non-triggering rainfall events. In my opinion, a larger inventory with more debris flow events and spanning a longer period is needed to provide a reliable calibration and verification of the proposed thresholds.

**Authors' reply:**

In response to the comments, it's important to note that the framework has been tested against one debris-flow event, but also four no-debris-flow events. The results demonstrated that the proposed framework has successfully predicted both the occurrence and non-occurrence of debris flow events. However, the authors acknowledge a significant limitation here application and testing of the framework were confined to a small catchment. This is related to the challenges associated with obtaining high-quality observed hydrological data in small and unstable channels. Additionally, the validation process relied on only one debris flow event, emphasizing a need for broader testing in similar catchments to enhance the framework's robustness. We have explicitly considered this limitation in the Discussion Section of the initial manuscript.

On the flip side, when compared to traditional statistical Intensity-Duration (I-D) analysis approaches, the proposed framework in this study offers a distinct advantage. It excels in generating rainfall thresholds for areas with limited historical data on debris flow occurrences. This makes the proposed framework particularly well-suited for regions where data is scarce. It is essential to highlight that in cases where a more extensive dataset is available, encompassing multiple debris flow events over an extended period, the traditional statistical I-D rainfall threshold method is recommended. This is due to its straightforward calculation process and the availability of influencing factors that can contribute to a more comprehensive analysis.

3. Finally, the structure of the paper needs to be improved. The manuscript lacks clear objectives. The results, methods and discussion are mixed through section 4 and in the discussion (section 5). Some information appears repeated, and some relevant

information to understand parts of the paper comes late. This makes it difficult for the reader

**Authors' reply:**

Thanks for the comments. The authors have revised the structure of the paper in the revised MS.

4. Line 691: I do not agree. In my opinion, statistical methods used to obtain empirical rainfall intensity-duration thresholds are objective. In fact, such thresholds are calibrated and validated using large datasets containing multiple landslide events and no-events and, in some cases, even monitoring data. The thresholds you proposed have not been properly validated using debris flow data.

**Authors' reply:**

The authors acknowledge the common practice of calibrating and validating Intensity-Duration (ID) statistical thresholds using extensive datasets that include multiple landslide events and non-occurrence events. These ID statistical thresholds are often recommended due to their straightforward calculation process. However, the authors express concern about the objectivity of the ID statistical methods, primarily arising from the absence of a universally accepted definition for rainfall events. The definition of a rainfall event plays a crucial role in establishing ID thresholds.

In the revised manuscript, the authors will revisit this statement to ensure a more accurate expression of such perspective on the objectivity of ID statistical thresholds is provided.

 The reviewer also could find the changes in Lines 683-685 of the revised MS.

5. Line 707: The approach was already presented in Wei et al., (2017).

**Authors' reply:**

Please see the response to Comment 1#.

**Responses to the Comments Raised by Reviewer #2**

1. Wei et al. explore the possibility of determining intensity-duration thresholds of runoff-induced debris flows through a combination of hydrological and hydrodynamic analysis. The authors present an alternative approach to deciphering the rainfall thresholds for debris flows instead of the traditional approach, which relies on statistical correlations of existing landslide information. I agree with the authors that this approach may be useful for areas having observational data on debris flow occurrences. The manuscript shows promising results. However, the authors have already presented similar works in two of their publications, i.e., Wei et al., 2017 and Wei et al., 2018, similarly on the same study area and using almost the same methods. Considering this, it is difficult to understand the original contribution of this study, the advancements, improvisation and improvements after the work of Wei et al., 2017 and Wei et al., 2018. Either the authors are not explaining it in detail, or I fail to understand where and how it is improved than the previous studies. In this regard, I recommend that the authors clearly present the advancements of this study compared to their previous publications.

**Authors' reply:**

Thank you for your comments. The framework presented in this study significantly diverges from our previous research. Firstly, in earlier studies led by Wei et al. (2018), the focus was solely on the hydrological model (NAM model), with the presentation of hydrological simulations. In Wei et al. (2017), a machine learning model was employed for runoff prediction. In contrast, our current study adopts an integrated hydrological and hydrodynamic modeling approach, providing a more robust estimation by directly incorporating overland flow dynamics in regions prone to debris flows.

The second key deviation lies in our approach to determining the occurrence of debris flow. In prior studies, such as Wei et al. (2018), peak discharge was used as the critical parameter. If the calculated peak discharge from the hydrological model exceeded the critical discharge, debris flow occurrence is confirmed; otherwise, it was deemed non-occurrence. However, in this study, we use hydrodynamic metrics as the critical parameter for predicting debris flow occurrences. This is in contrast to the common practice, such as our earlier works (Wei et al., 2018 and Wei et al., 2017), where peak

discharge served as the critical parameter. Now peak discharge has become widely accepted as the standard critical parameter for predicting debris flow occurrences. For instance, Li et al. (2021) established rainfall intensity-duration thresholds based on process-based critical runoff discharge. Bernard and Gregoretti (2021) proposed determining debris flow occurrences through a hydrological model coupled with a critical discharge relationship using rainfall and raw radar data. However, it is important to note the limitations of the previous frameworks, which solely relied on peak discharge predicted only by a hydrological model. Such an approach could predict the occurrence of debris flow but lacked the capability to predict the scales of debris flow. In essence, our previous model could only forecast the likelihood of debris flow occurrence, without providing insights into the magnitude of the runoff-generated debris flow.

In this study, we introduce a novel framework that employs an integrated hydrological and hydrodynamic modeling approach to enhance the accuracy of rainfall thresholds estimation for runoff-generated debris flows. The incorporation of a hydrodynamic model enables the prediction of detailed flow dynamics, providing grid-based information such as water depth and flow velocity in regions susceptible to debris flows. The flow information obtained is then employed to compute hydrodynamic metrics based on unit-width discharge. These metrics are compared with corresponding hydrodynamic thresholds, serving as indicators for the occurrence of runoff-generated debris flows. Given that the derived hydrodynamic indices are grid-based, the framework allows for determining the proportion of trigger cells within the triggering area or the total number of trigger cells. This leads to clear advantages of the current framework against existing approaches, enabling not only the prediction of debris flow occurrences but also providing insights into the magnitude and scales of debris flow. As a summary, in contrast to our previous study and similar works, such as Li et al. (2021), our approach represents a significant advancement in the prediction of runoff-generated debris flows.

Furthermore, to evaluate debris flow occurrences at the catchment scale, we have introduced a new concept as "zone threshold." The zone threshold is defined as the

critical proportion of trigger cells within the triggering area. If the proportion of trigger cells surpasses the zone threshold, it signifies a debris flow occurrence; otherwise, it is categorized as a non-occurrence event. This concept integrates zone thresholds with hydrodynamic thresholds within the proposed framework. The advantage of incorporating zone thresholds lies in their ability to represent varying degrees of conservatism or adventurousness in rainfall thresholds. A smaller zone threshold corresponds to a lower rainfall threshold, reflecting a more conservative approach. Conversely, a larger value results in a higher rainfall threshold, indicating a more adventurous approach. By associating different zone thresholds with corresponding levels of warning, the framework facilitates decision-making and response actions based on identified rainfall thresholds. This approach allows for a spectrum of risk management strategies tailored to the desired level of caution or preparedness.

Overall, with the integration of the hydrodynamic model, the introduction of a critical parameter, and the incorporation of zone thresholds, we assert that our study makes a novel contribution to the prediction of runoff-generated debris flows.

The Reviewer also could find the changes in Lines 93-99, Lines 105-113 and Lines 697-725 of the revised MS.

*References:*

*Wei, Z.L., Shang, Y.Q., Zhao, Y., Pan, P. and Jiang, Y.J., 2017. Rainfall threshold for initiation of channelized debris flows in a small catchment based on in-site measurement. Engineering Geology, 217:23-34.*

*Wei, Z.L., Xu, Y.P., Sun, H.Y., Xie, W., Wu, G., 2018. Predicting the occurrence of channelized debris flow by a cascading flood debris-flow model in a small debris flow-prone catchment. Geomorphology, 308 :78-90*

*Li, Y.J., Meng, X.M., Guo, P., Dijkstra, T., Zhao, Y., Chen, G., Yue, D.X., 2021. Constructing rainfall thresholds for debris flow initiation based on critical discharge and S-hydrograph. Engineering Geology, 280:105962.*

*Bernard, M. and Gregoretti, C. 2021. The use of rain gauge measurements and radar data for the model-based prediction of runoff-generated debris-flow occurrence in early warning systems. Water Resources Research, 57(3): e2020WR027893.*

The Reviewer also could find the changes in Lines 105-113 and Lines 697-725 of the revised MS.

2. On the other hand, the approach adopted by the author is also conceptualized by van Asch et al. 2014; Domènech et al., 2019; Siva Subramanian et al., 2023. It would be interesting for the readers to know the difference between the approaches of these studies and those of the current manuscript.

**Authors' reply:**

The authors acknowledge the similarity in conceptualization between this study and the mentioned research. Both studies aim to establish Intensity-Duration (ID) rainfall thresholds through a numerical approach. The mentioned studies build on the numerical framework proposed by van Asch et al. (2014). In those studies, erosion by runoff is considered to occur when the bed shear stress ($\tau$, kPa) exceeds the critical erosive shear stress at the initiation of soil erosion ($\tau c$, kPa), and the volumetric concentration of solids in the debris flow (Cv) is smaller than an equilibrium value (Cv∞).

In contrast, our study diverges in its approach to determining debris flow occurrences. We focus more on the values of the hydrodynamic index (unit-width discharge) at each cell. When the calculated unit-width discharge exceeds the critical threshold, a debris flow is considered to happen. This establishes a distinct criterion/approach for determining debris flow occurrence compared to the mentioned studies.

The current study focuses on runoff-generated debris flows, where the most significant triggering factor is surface flow dynamics. The transition from clear water flow to debris flow hinges on hydrodynamic conditions, such as flow discharge, surpassing certain thresholds. Therefore, providing accurate spatially distributed hydrodynamic information, including water depth and velocity, is crucial for estimating the occurrence of runoff-generated debris flows. This differs significantly from the initiation of landslide-triggered debris flows, where infiltration plays a pivotal role. In the mentioned studies, the hydrological simulation is simplified, calculating runoff using a basic lumped infiltration model that neglects the initial moisture content of the soil. This underscores that the framework proposed in this study is tailored specifically for runoff-generated debris flows, given its primary focus on the hydrological processes associated with such events.

The Reviewer also could find the changes in Lines 697-705 of the revised MS.

3. I also request that the authors kindly use a considerate tone while referring to approaches that vary from the author's perspective i.e., statistical approaches.

**Authors' reply:**

Thanks for pointing this out. In the revised manuscript, we will use an appropriate tone to discuss and comment on other methodologies that differ from our perspective.

4. Line 123: Figure 1. Please revise this figure as a flow chart. The intensity-duration threshold curve looks unrealistic in shape. Please verify.

**Authors' reply:**

Thanks for the comments. Figure 1 serves as a conceptual illustration of the framework. Therefore, the I-D threshold curve depicted in the figure is not intended to be a realistic representation at this stage. We attempted to revise Figure 1 as a flow chart; however, the resulting height of the figure would be too large. Therefore, we have decided to maintain Figure 1 in its current form.

5. Line 356: Figure 5. Is this rainfall vs observed discharge from a debris flow event? Please explain whether the instrumentation and calibration using the NAM model will apply during an actual debris flow.

**Authors' reply:**

In Fig. 5, it's important to note that there are no debris flow events during the depicted rainfall events. The observed discharge in Fig. 5 corresponds to clear water flow.

The Reviewer also could find the changes in Lines 362-364 of the revised MS.

6. Line 421: How does this curve appear during an actual debris flow? The runoff values will be within the said range or higher? This question comes because it is unclear whether the approach actually simulates the erosion caused by runoff or only the runoff. Please explain.

**Authors' reply:**

The simulated discharge in Fig. 6 represents clear water flows. We anticipate that during an actual debris flow event, the discharge for this curve will significantly increase. This expectation aligns with the fact that the volume of a debris flow is typically several times larger than a clear water flow.

The Reviewer also could find the changes in Lines 419-420 of the revised MS.

---

## Author Response (AR2)

**Revision notes on Manuscript No. NHESS-2023-180**

First of all, the authors thank the Editors and the reviewers for considering our manuscript and providing constructive comments to help us improve the quality of our work. We have accordingly revised the manuscript by carefully addressing or answering the comments point-by-point, summarized as follows. Following the revision, we hope we have clarified all of the points summarized by the Editor and reviewers.

**Responses to the Comments Raised by Reviewer #1**

1. I appreciate the authors' efforts in addressing some of my comments. They have significantly improved the structure of the paper and provided a better explanation of the novelties in their work. However, the objectives are still not clearly stated. Lines 100-111 are more an anticipation of the conclusions.

**Authors' reply:**

Thank you for your comments. The authors have changed the content of Line 100-111 as below and moved the content of Line 100-111 to the Conclusion part of the revised MS.

*"To fill the current research and practical gaps, we aim to propose in this study a new framework based on an integrated hydrological and hydrodynamic modeling approach to more reliably estimate the rainfall thresholds of runoff-generated debris flows, i.e. providing a physically based approach to estimate trigger-cause-based rainfall thresholds. In addition, the proposed modeling framework will effectively incorporate meteorological conditions, catchment topographic properties, and grain-size distribution of debris materials, making it more suitable for application in areas with limited historical data. The rest of the paper is organized as follows: Section 2 describes the proposed framework; Section 3 introduces the case study including the flow monitoring scheme; Section 4 presents the validation results; Section 5 discusses the advantages and limitations of the proposed method, followed by brief conclusions drawn in Section 6".*

The reviewer can also find the changes in Lines102-111 and Lines 790-797 in the revised MS.

2. The authors are still overstating their conclusions, which have been drawn from a generalization based on limited data (e.g. lines 505-506, 515-520, 628-630). The authors should either provide more proof or tone down their claims.

**Authors' reply:**

Thanks for the comments. We fully agree with the reviewer that some concluding statements are overstated. We have revised the sentences of Lines 505-506, 515-520, 628-630, as follows:

Line 505-506

*The results are consistent with the actual observation, i.e. a debris flow did occur during the typhoon event, demonstrating that the proposed methodology successfully predicted the occurrence of this debris flow event.*

The reviewer can also find the changes in Lines 504-506 in the revised MS.

Line 515-520

*This implies that the hydrodynamic conditions necessary for triggering a debris flow are met only in a small fraction of the grid cells and the likelihood of debris flow occurrence is low. This conclusion aligns with the actual observations, i.e. no debris flow was observed during these six rainfall events. As a whole, these numerical tests demonstrate the capability of the framework including the adopted hydrodynamic thresholds in predicting six observed non-debris flow events and one actual debris flow event.*

The reviewer can also find the changes in Lines 517-520 in the revised MS.

Line 628-630

*From Fig. 14, it is evident that both the proposed I-D rainfall threshold and the empirical rainfall threshold can effectively distinguish one triggering and six non-triggering rainfall events, highlighting the feasibility of the proposed framework.*

The reviewer can also find the changes in Lines 650-653 in the revised MS.

3. The accumulation recorded during the October debris flow (around 300 mm) and the accumulation registered during the non-triggering rainfalls shown in Table 7 (between 16 mm and 60 mm) are very different, and therefore the resulting discharges are also very different. These rainfall events have been used to decide the % of catchment area that has to exceed the hydrodynamic thresholds to trigger a debris flow. I would expect that rainfall with larger accumulation than those 6 non-triggering rainfalls would also result in larger discharges, which probably result in larger areas exceeding the hydrodynamic thresholds. How sure are you that such events would not trigger a debris flow in the catchment? Do you have any additional examples of rainfall events registered between 2013 and today that have not triggered debris flows in the studied catchment that could be used for further validation and calibration of the zone thresholds?

**Authors' reply:**

Due to the failure to obtain high-quality observed hydrological data in the small and unstable channels, the monitoring station has not been maintained and has since been out of work after September 2013. So, we, unfortunately, do not have additional data for both occurred and non-occurred events between 2013 and today to further validate the zone thresholds in the study area. Due to the lack of data, the authors proposed rainfall thresholds for the study area with different assumed zone thresholds (see Fig. 15 in revised MS). Sensitivity analysis on the rainfall thresholds regarding the zone thresholds was also conducted. The authors acknowledge such a major limitation in this study, as the application and testing of the framework were limited to a small number of events in a small catchment. This highlights the need for broader testing in similar catchments to enhance the robustness of the framework. This limitation was explicitly discussed in the Discussion Section of the original manuscript. This is actually the challenge we would like to address in this study, i.e. by proposing a framework based on physical principles, which is more suitable for areas with limited historic debris flow data. The main objective of this manuscript is to explore the feasibility of a coupled hydrological and hydrodynamic modeling framework for estimating rainfall thresholds for debris flow occurrence. Due to the limited availability of the observation data, we

have used one debris flow event and six non-debris flow events to validate the proposed framework, but not to determine the values of zone thresholds. Since it is a physically based approach, the users may propose their own rainfall thresholds within the framework if they can establish the relationship between rainfall and runoff and given corresponding values for zone thresholds. Actually, the selection of zone thresholds should vary depending on the definition of debris flow occurrence. For example, choosing a 10% zone threshold means that debris flow occurrence is considered positive when 10% of the computed domain is identified as triggering cells. Choosing a 15% threshold means that debris flow occurrence is considered positive when 15% of the computed domain is identified as triggering cells. Such a threshold may be more reliably defined when data is available for different applications, and the selection of zone thresholds is linked to varying degrees of conservatism or adventurousness in rainfall thresholds. A smaller zone threshold corresponds to a lower rainfall threshold, reflecting a more conservative approach, while a larger value indicates a higher rainfall threshold, suggesting a more adventurous approach. Different zone thresholds correspond to different levels of warning, with smaller values corresponding to lower warning levels and larger values to higher warning levels. So, we recommend that the selection of zone thresholds for a specific catchment should be conducted based on risk management decisions and formally established by governmental authorities in advance. This approach allows for the implementation of a variety of risk management strategies that can be customized to meet the desired level of caution or preparedness.

4. The same triggering rainfall, and non-triggering rainfall events that have been used to select the zone thresholds have been used to check the performance of the proposed thresholds. This is an important limitation. The recorded discharges of such events are significantly different, and of course it is possible to separate very well between the events and no-events (Fig 13 and Fig 14). Do you have proof that all rainfall events resulting in larger discharges than the one that triggered the debris flow in 2013 would also result in debris flow events? Similarly, what about rainfall events that result in slightly lower discharges than the rainfall that triggered the debris flow, are you sure

any of them could trigger a debris flow?

**Authors' reply:**

In this manuscript, one triggering debris flow event and six non-triggering debris flow events were utilized to validate the effectiveness of the proposed framework, rather than to determine the zone thresholds, which should be user-defined with more observations available and for different applications. The rainfall thresholds presented in Figures 13 and 14 were established based on various design hyetographs generated from Intensity-Duration-Frequency (IDF) analysis, rather than from observed rainfall events. This implies that the rainfall events used to assess the performance of the proposed thresholds differ from those used to develop them. It's noteworthy that the discharge calculated from some design hyetographs is notably consistent (see Table 1), in contrast to the variability observed in recorded discharge data.

Table1 the peak discharge under different IDF analysis

| IDF | 1h-100y | 3h-20y | 1h-10y | 3h-5y | 3h-3y | 1h-5y |
|---|---|---|---|---|---|---|
| Peak discharge($m^3$/s) | 0.38 | 0.37 | 0.16 | 0.17 | 0.12 | 0.11 |

To further demonstrate the feasibility of the threshold framework, we have also conducted several extra scenario simulations. Specifically, we modified the input rainfall events, varying the cumulative amount from 10% to 110% of the original rainfall that triggered the 2013 debris flow. The rainfall distribution is presented in Fig. 1. The cumulative input rainfall ranges from 38 mm to 418 mm, with the original value that triggered the 2013 debris flow being 380 mm. The calculated discharge for each scenario is shown in Fig. 2. Additionally, the calculated peak discharge and the corresponding proportion of triggering cells based on the Thresholds G are detailed in Table 2.

[Figure]

Fig.1 Rainfall distribution of the scenario simulations.

[Figure]

Fig.2 Calculated runoff of the scenario simulations.

Table 2 Calculated peak discharge and the corresponding proportion of triggering cells based on Threshold G

| Cumulative amount of input rainfall (mm) | 418 | 342 | 285 | 190 | 95 | 38 |
|---|---|---|---|---|---|---|
| Calculated peak discharge (m³/s) | 2.2 | 1.7 | 1.3 | 0.75 | 0.25 | 0.048 |
| Proportion of triggering cells (%) | 48 | 44 | 39 | 33 | 17 | 3 |

The calculated peak discharge from the rainfall that triggered the 2013 debris flow is 2.0 m³/s. As shown in Table 2, rainfall events resulting in discharges greater than the one in 2013 could also result in debris flow events, with the calculated proportion of triggering cells for a 418 mm rainfall event being 48%. Even slightly lower discharges than the 2013 event may trigger a debris flow, as indicated by the 342 mm rainfall event, which has a calculated triggering cell proportion of 44%. This demonstrates that the calculated proportion of triggering cells is consistent with the change in the input rainfall.

The reviewer can also find the changes in Lines 526-549 in the revised MS.

5. In line 680, it is stated that the reliability of statistical I-D thresholds depends on the data being used and that long-term observations are essential to derive reliable thresholds. I think your approach also depends on the quality of the data being used to obtain the thresholds. To be able to select adequate and objective zone thresholds, you also need to have a long time series of observations of debris flow events and no-events.

**Authors' reply:**

We fully agree with the reviewer that a long time series of observations is essential to validate the true values of the zone threshold of a specific catchment. However, as mentioned in Comment 3, the selection of zone thresholds depends on the definition of debris flow occurrence. These specific values are not predetermined for individual catchments. Therefore, to calculate the rainfall thresholds based on the proposed framework, governmental decision-makers or other users must assign the values for zone thresholds according to applications. Consequently, the zone threshold should be established by users to reflect catchment settings and applications. Whilst it is important

to show the feasibility of the approach, it is not entirely essential to be validated by multiple debris flows or non-debris flow events in the current case study.

The reviewer can also find the changes in Lines 748-752 in the revised MS.

6. Line 66-67: In the literature you can find a number of examples (e.g. Oorthuis et al., 2023, Hirschberg et al., 2021, Bel et al., 2017, Abancó et al., 2016), of reliable statistical rainfall thresholds defined for specific catchments where monitoring data exists (and thus a complete register of debris flow occurrences).

**Authors' reply:**

The authors fully agree that reliable statistical rainfall thresholds for a specific catchment could be derived when an adequate dataset of debris flow events is available. The authors have revised the sentences as below:

*"So, the reliable statical rainfall thresholds for a specific catchment may be derived when sufficient high-quality data of debris flow events is available (Oorthuis et al., 2023, Hirschberg et al., 2021, Bel et al., 2017, Abancó et al., 2016). However, in the areas with limited data availability, it will be technically challenging to reliably define the statistical I-D thresholds of debris flows. It may be more useful to propose a physical-based approach to estimate the rainfall thresholds in such data-scarce area".*

The reviewer can also find the changes in Lines 66-71 in the revised MS.

7. Line 475, and section 4.3: What is the return period of the October 2013 rainfall event? How long did the event last?

**Authors' reply:**

The total rainfall amount of October 2013 rainfall event is 380 mm with the rainfall duration being 16 hours. The rainfall conditions of the Typhoon Fitow (October 2013) event are also included in Fig 15 of revised MS, which has a return period of about 100 years.

The reviewer can also find the changes in Lines 471-472 in the revised MS.

8. Line 505-506: It proves that you can distinguish this debris-flow event.

**Authors' reply:**

The authors have revised the sentence as below:

*The results are consistent with the actual observation, i.e. a debris flow did occur during the typhoon event, demonstrating that the proposed methodology can be used to predict the occurrence of such an event.*

The reviewer can also find the changes in Lines 504-506 in the revised MS.

9. Line 629: The analysed 6 non-triggering rains and 1 triggering rain.

**Authors' reply:**

The authors have revised the sentence as below:

*From Fig. 14, it is evident that both the proposed I-D rainfall threshold and the empirical rainfall threshold can effectively distinguish the one triggering and six non-triggering rainfall events, highlighting the feasibility of the proposed framework.*

The reviewer can also find the changes in Lines 650-653 in the revised MS.

10. Line 680: Clearly is a very strong word.

**Authors' reply:**

The authors have revised the sentence as below:

*This indicates that a statistical approach may be difficult to be reliably applied in ungauged catchments where high-quality historical data is missing.*

The reviewer can also find the changes in Lines 706-708 in the revised MS.

---

## Author Response (AR3)

<h1 style="text-align:center">Revision notes on Manuscript No. NHESS-2023-180</h1>

First of all, the authors thank the Editor for considering our manuscript and providing constructive comments to help us improve the quality of our work. We have accordingly revised the manuscript by carefully addressing or answering the comments point-by-point, summarized as follows. Following the revision, we hope we have clarified all of the points summarized by the Editor.

**Responses to the Comments Raised by Editor**

1. I have reviewed your manuscript and noted that it has been significantly improved by addressing many of the points raised by the referees. However, I believe the conclusions should better highlight the limitations of the work. As one of the referees mentioned, it is important to call for further measurements to validate the proposed approach. Although the approach is designed for data-scarce cases, the more data available for validation, the more robust the assessments of the model's performance. Currently, you use only a few events for this purpose, so additional studies should be conducted to better assess the validity of the proposed model.

**Authors' reply:**

Thank you for your positive comments. The authors have revised the conclusions and highlighted the limitations of our work as below:

*"However, there are still some limitations of this work. The main limitation is that the proposed framework was tested on only one debris flow event and a few non-debris flow events. Further measurements are needed to validate the proposed approach comprehensively. Although the approach is designed for data-scarce cases, having more data available for validation will make the assessments of the model's performance more robust. Therefore, additional studies should be conducted in similar catchments to better evaluate the validity and reliability of the proposed model in the future.*

The Editor can also find the changes in Lines 797-803 in the revised MS.

2. Additionally, the figures should be improved to ensure consistency (e.g., same font

type and size for all figures).

**Authors' reply:**

Thank you for the comments. We have revised the figures in the manuscript to ensure consistency, adjusting the font type and size to be uniform across all figures. The Editor can also find the changes in the revised MS.